

# $T\overline{T}$-like flows and $3d$ nonlinear supersymmetry

**Christian Ferko**[1*]**, Yangrui Hu**[2†]**, Zejun Huang**[3‡]**,
Konstantinos Koutrolikos**[4°] **and Gabriele Tartaglino-Mazzucchelli**[3§]

**1** Center for Quantum Mathematics and Physics (QMAP), Department of Physics &
Astronomy, University of California, Davis, CA 95616, USA
**2** Perimeter Institute for Theoretical Physics, Waterloo, ON N2L 2Y5, Canada
**3** School of Mathematics and Physics, University of Queensland,
St Lucia, Brisbane, Queensland 4072, Australia
**4** Department of Physics, University of Maryland, College Park, MD 20742-4111, USA

★ caferko@ucdavis.edu , † yhu3@perimeterinstitute.ca , ‡ zejun.huang@uq.net.au ,
° koutrol@umd.edu , § g.tartaglino-mazzucchelli@uq.edu.au

## Abstract

We show that the $3d$ Born-Infeld theory can be generated via an irrelevant deformation of the free Maxwell theory. The deforming operator is constructed from the energy-momentum tensor and includes a novel non-analytic contribution that resembles root-$T\overline{T}$. We find that a similar operator deforms a free scalar into the scalar sector of the Dirac-Born-Infeld action, which describes transverse fluctuations of a D-brane, in any dimension. We also analyse trace flow equations and obtain flows for subtracted models driven by a relevant operator. In $3d$, the irrelevant deformation can be made manifestly supersymmetric by presenting the flow equation in $\mathcal{N} = 1$ superspace, where the deforming operator is built from supercurrents. We demonstrate that two supersymmetric presentations of the D2-brane effective action, the Maxwell-Goldstone multiplet and the tensor-Goldstone multiplet, satisfy superspace flow equations driven by this supercurrent combination. To do this, we derive expressions for the supercurrents in general classes of vector and tensor/scalar models by directly solving the superspace conservation equations and also by coupling to $\mathcal{N} = 1$ supergravity. As both of these multiplets exhibit a second, spontaneously broken supersymmetry, this analysis provides further evidence for a connection between current-squared deformations and nonlinearly realized symmetries.

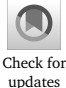

# 1   Introduction

A very useful tool for the exploration of the space of theories within a specific theoretical framework is that of *flows*. This tool has been used extensively in physics and mathematics with enormous success. The most famous application of a flow in theoretical physics is the renormalization group (RG) flow which is used to explore the space of field theories. In mathematics, a well-known example is that of Ricci flow [1] which explores the space of Riemannian manifolds. Specifically, RG flows run from high energy (UV) theories to low energy (IR) theories and determine the set of coupling constants of QFTs along the trajectory of the flow in a manner controlled by the beta function. Similar in spirit, Ricci flows create a trajectory in the space of Riemannian manifolds, dictated by the Ricci tensor, and determine the metric of the manifolds along the trajectory. Both flows have interesting points. RG flows have fixed points that correspond to conformal field theories and Ricci flows have attractors which correspond to constant curvature manifolds.

The success of RG flows suggests further exploration of the space of QFTs using additional flows triggered by operators that deform theories in new and interesting ways. A special example is the $T\overline{T}$ operator, which is irrelevant in the sense of the RG.[1] Ordinarily, turning on an irrelevant operator subsequently requires the addition of infinitely many counterterms, and thus leads to a loss of analytic control. Nonetheless it has been shown [2,3] that the $T\overline{T}$

---

[1]The coupling constant that turns on and controls the contribution of this operator has negative mass dimension. Such a coupling is irrelevant at low energies but grows more important at high energies.

operator in two dimensions is an exception to this rule, and in fact the deformation is "solvable" in the sense that several quantities in the deformed theory can be computed analytically in terms of the data of the undeformed theory.

The definition of the $T\overline{T}$ operator is historically given (up to normalisation factors) as

$$T\overline{T} := -\det(T_{mn}) = \frac{1}{2}\left(T^{mn}T_{mn} - T^m_{\ m} T^n_{\ n}\right),\tag{1}$$

where $T_{mn}$ is the energy-momentum tensor of the system. In $2d$, using the holomorphic and anti-holomorphic coordinates, this operator takes the form $T\overline{T} \sim T_{zz}\overline{T}_{\bar{z}\bar{z}} - T_{z\bar{z}}^2$. For CFTs, the trace part ($T_{z\bar{z}}$) vanishes and we are left only with the $T_{zz}$ and $\overline{T}_{\bar{z}\bar{z}}$ term, hence the name $T\overline{T}$.[2]

There are various viewpoints that provide intuition as to why $T\overline{T}$ deformations are not the typical irrelevant operators and should be studied in more detail. One such viewpoint is its property of being an integrable deformation. This means that if we start with an integrable theory and follow its flow under $T\overline{T}$, then the output will also be an integrable theory [8,9]. In other words, the preservation of integrability by the flow makes $T\overline{T}$ very special. A different viewpoint, not related to integrability, involves connections between the $T\overline{T}$ deformation and gravity. For instance one can calculate the contribution of an infinitesimal $T\overline{T}$-deformation to the partition function of the theory and show that it is equivalent to an integration over random variations of the underlying geometry [10]. A second connection involves coupling to flat space JT gravity [11,12], which has more recently been understood as a special case of a topological gauging procedure that can be applied to more general current-current deformations [13]. A third approach is to geometrize $T\overline{T}$ and related deformations via a dynamical change of coordinates which couples the undeformed theory to a field-dependent background metric [14–18].

Unlike RG flows, $T\overline{T}$ flows run towards higher energies. Nevertheless there are examples of quantities that have been computed exactly using this flow. In this paper we will focus on a specific quantity, the deformed classical Lagrangian. This means that we: (*i*) consider the space of field theories that have a Lagrangian description $\mathcal{L}$, (*ii*) assume a curve through this space parametrized by some parameter $\lambda$ such that the point on the curve corresponding to value $\lambda = \lambda_0$ is the field theory with Lagrangian $\mathcal{L}^{(\lambda_0)}$ and (*iii*) define the operator $\mathcal{O}^{(\lambda)}$ that triggers the flow from point $\lambda$ to point $\lambda + \delta\lambda$ as

$$\mathcal{L}^{(\lambda+\delta\lambda)} = \mathcal{L}^{(\lambda)} + \delta\lambda\, \mathcal{O}^{(\lambda)}.\tag{2}$$

For the case of $T\overline{T}$-flow of Lagrangians in $2d$ we consider the following deformation:

$$\frac{\partial \mathcal{L}^{(\lambda)}}{\partial \lambda} = \det\left(T_{mn}^{(\lambda)}\right).\tag{3}$$

A well-known example of solving the $T\overline{T}$-flow equation for the deformed Lagrangian is the case of a free boson in two dimensions: $\mathcal{L} = \partial\phi\overline{\partial}\phi$. In [3] it was shown that under a $T\overline{T}$-flow this theory is mapped to the $3d$ Nambu-Goto action in the static gauge:

$$\mathcal{L}^{(0)} = \partial\phi\overline{\partial}\phi \;\rightarrow\; \mathcal{L}^{(\lambda)} = \frac{1}{2\lambda}\left(\sqrt{4\lambda\partial\phi\overline{\partial}\phi + 1} - 1\right) = -\frac{1}{2\lambda} + \mathcal{L}_{NG}.\tag{4}$$

An interesting feature of the above deformed Lagrangian is the square root. Usually this type of nonlinearity *(a)* corresponds to the resummation of infinitely many terms, which in this case can be thought of as being generated by iteratively adding and re-computing the controlled

---

[2]However, this name is somewhat misleading since a $T\overline{T}$-deformed CFT at finite deformation parameter is no longer conformally invariant and thus its stress tensor has non-vanishing trace. Interestingly, it appears that $T\overline{T}$-deformed CFTs exhibit an unconventional field-dependent conformal symmetry [4–7].

$T\overline{T}$ combination along the flow, and *(b)* appears in many attractive theories. In this example, the Nambu-Goto action describes the propagation of bosonic strings with the flow parameter $\lambda$ playing the role of inverse string tension.

A natural question to ask is whether other "square root" Lagrangians have a similar interpretation: can they be obtained from a $T\overline{T}$-like deformation of a free theory? Another well-known and extensively studied member of the square root Lagrangians is the Born-Infeld (BI) theory and its generalizations. The BI Lagrangian provides a nonlinear extension of Maxwell theory, but most importantly it is the leading term in the low-energy effective description of D-branes [19,20]. Because of this, BI-type actions beautifully capture a solitonic realization of the partial supersymmetry breaking phenomenon [21–26].

This can be understood using some basic properties of the theory. As solitonic solutions of open string theory, D-branes introduce boundaries that break the translational symmetries in their perpendicular directions. However, because translational symmetries are the source of supersymmetry, fewer translational symmetries lead to less supersymmetry. In fact, based on the size of spinors it is straightforward to see that for every translational symmetry lost, the number of supercharges of the theory is cut in half.

Nevertheless, it is a fact of physics that the memory of broken symmetries is never lost. The theory still realizes these symmetries but in a nonlinear manner. Hence, the nonlinearities of BI Lagrangians may be interpreted as a manifestation of partial supersymmetry breaking and the existence of a Goldstino mode. It would be very interesting to investigate whether the same nonlinearities can also be generated by a $T\overline{T}$-like flow. This has been shown to be true for certain models in two dimensions up to $\mathcal{N}=(2,2)$ supersymmetry as well as $\mathcal{N}=1$, $4d$ theories, see [27–29]. This suggests that non-linearly realized (super)symmetries might also arise from $T\overline{T}$-like flow equations, at least when the seed theories possess extra symmetries such as the shift-symmetries of free models.

In this paper, we consider the $3d$, $\mathcal{N}=1$ supersymmetric Born-Infeld theory [30–32] which is related to the effective description of D2-branes in type IIA string theory. This theory was obtained by partially breaking supersymmetry from $\mathcal{N}=2$ to $\mathcal{N}=1$ in such a manner that the supersymmetric multiplet which manifests the surviving, linearly realized, supersymmetry remains massless (i.e. it is the Goldstone multiplet). It was found that the role of the Goldstone multiplet can be played by either the $3d$, $\mathcal{N}=1$ vector multiplet or its dual tensor multiplet.

This work aims to show that these results can also be obtained from a $T\overline{T}$-like flow. The deformation parameter which labels points along the flow trajectory is related to the scale of supersymmetry breaking and more precisely to the VEV ($\kappa$) of the auxiliary $\mathcal{N}=1$ superfield of the $\mathcal{N}=2$ supermultiplet. The flow relates the $\kappa \to \infty$ limit of these theories, which corresponds to the manifestly supersymmetric Lagrangians that describe the $3d$, $\mathcal{N}=1$ free Maxwell and free tensor multiplet, to theories at finite (but not zero) values of $\kappa$, which describe the $3d$, $\mathcal{N}=1$ supersymmetric Born-Infeld theory in terms of the Maxwell-Goldstone and tensor-Goldstone multiplets, respectively.

As a result, the $3d$ BI Lagrangian – which is the bosonic truncation of the above supersymmetric theories – arises from a $T\overline{T}$ deformation. This may come as a surprise since it was explicitly checked in [33] that a flow equation of the form

$$\frac{\partial \mathcal{L}}{\partial \lambda} = c_1 T^{ab} T_{ab} + c_2 \left( T^a_{\ a} \right)^2 , \tag{5}$$

does not lead to Born-Infeld type solutions in any dimension other than $d = 4$, regardless of the choice of coefficients $c_1$ and $c_2$. To evade this no-go result, one must introduce some additional ingredient. We will find that the necessary addition is a new operator $R$ which is

non-analytic in the energy-momentum tensor:

$$R = \sqrt{\frac{1}{d} T^{ab} T_{ab} - \frac{1}{d^2} \left( T^a_{\ a} \right)^2} = \frac{1}{\sqrt{d}} \sqrt{\widehat{T}^{ab} \widehat{T}_{ab}} \,,$$

$$\widehat{T}_{ab} = T_{ab} - \frac{1}{d} g_{ab} T^c_{\ c} \,. \tag{6}$$

This operator $R$ is constructed from the traceless part $\widehat{T}_{ab}$ of the stress tensor and is the $d$-dimensional analogue of the root-$T\overline{T}$ operator whose two-dimensional version was studied in [34] (see also [35–42]).[3] A marginal flow driven by the operator $R$ can be used to obtain the Modified Maxwell (ModMax) theory, which was introduced in [44–46], in four spacetime dimensions [47]. This root-$T\overline{T}$-like flow can also be made manifestly supersymmetric [48], and along with the 4$d$ $T\overline{T}$-like flow, forms a 2-parameter family of commuting deformations which generate the ModMax-Born-Infeld theory (or its supersymmetric extension, in the superspace case) [49]. In two dimensions, the analogous root-$T\overline{T}$ flow can be used to construct a "Modified Scalar" theory, or combined with $T\overline{T}$ to yield a 2-parameter family of "Modified-Nambu-Goto" theories [17, 34, 50]. Classical root-$T\overline{T}$ flows in 2$d$ seem to enjoy some of the special properties of the ordinary $T\overline{T}$ deformation, such as preserving integrability in certain examples [51], although it is not yet known whether this deformation can be defined quantum-mechanically.

In the present work, we will not consider such marginal flows driven by the root-$T\overline{T}$-like operator $R$. Instead, we will only use $R$ as a tool to construct more general *irrelevant* flows. More precisely, we will expand the class of $T\overline{T}$-like flow equations (5) by including a third term $c_3 T^a_{\ a} R$ on the right side. A deformation which includes this additional term will be sufficient to generate the 3$d$ Born-Infeld theory, or its dual scalar theory, via a $T\overline{T}$-like flow, and likewise to generate the supersymmetric extensions of these theories by appropriate flows driven by supercurrents.[4]

The paper is organized as follows. In Section 2 we focus on the bosonic truncations of the supersymmetric D2-brane actions of interest. We present a flow driven by an irrelevant operator which deforms the 3$d$ free Maxwell theory to the Born-Infeld theory, which is the bosonic part of the Maxwell-Goldstone multiplet. We also show that a similar flow connects the 3$d$ free scalar to the scalar sector of the Dirac-Born-Infeld action, which is a truncation of the tensor-Goldstone multiplet, and that an analogue of this result holds in any spacetime dimension $d$. We then analyse and extend the trace flow equations presenting new classes of 3$d$ relevant flows for subtracted Lagrangians. In Section 3 we turn to the supersymmetriza-tion of these results. We view the results of [30] as a flow that connects 3$d$ super-Maxwell to supersymmetric Born-Infeld and we define the appropriate superspace operator $\mathcal{O}_{\kappa^2}$ trig-gering the flow. Furthermore, we explicitly construct the supercurrent multiplet for these theories by solving the corresponding superspace conservation equation. This is the multiplet that contains the energy-momentum tensor and its superpartner, the supersymmetry current. The supercurrent and supertrace superfields that define the supercurrent multiplet are then used to establish appropriate $T\overline{T}$-like superspace operators $\mathcal{O}_{T^2}$, $\mathcal{O}_{\Theta^2}$, $\mathcal{O}_{\Theta R}$ with the property that their component expansion includes terms proportional to $\widehat{T}^{ab} \widehat{T}_{ab}$, $(T^a_{\ a})^2$ and $(T^a_{\ a})R$ respectively. We show that for both descriptions of the 3$d$ supersymmetric BI theory —i.e. the Maxwell-Goldstone or the tensor-Goldstone multiplets— the superspace flow operator $\mathcal{O}_{\kappa^2}$ can be expressed uniquely as a linear combination of the superfields $\mathcal{O}_{T^2}$, $\mathcal{O}_{\Theta^2}$, $\mathcal{O}_{\Theta R}$ and thus identify this flow as a $T\overline{T}$-like flow. These flows are the manifestly supersymmetric extensions

---

[3]Although there have been some proposals [10,43] for $T\overline{T}$-like deformations in higher dimensions which involve roots of the determinant of the stress-energy tensor, such as $[\det(T)]^{1/(d-1)}$, note that $R$ is not proportional to any power of the determinant of $T_{ab}$.

[4]For other work on $T\overline{T}$ and supersymmetry, see [52–61].

of those in Section 2. Indeed we check that the bosonic truncation of the superspace results of this section is consistent with the bosonic flows of the previous section. In Section 4, we derive the above-mentioned supercurrents by coupling a large class of vector and tensor/scalar models to $\mathcal{N} = 1$ supergravity. Finally, in Section 5 we summarize these results and identify a few directions for future investigation. We have also collected our conventions and details of some calculations in Appendices A and B. An alternate presentation of one of our bosonic flows is presented in Appendix C.

## 2 Bosonic flows

In this section, we will truncate the D2-brane effective actions of interest to the bosonic degrees of freedom and study stress tensor flows for the resulting theories in components. This analysis provides a warm-up for the manifestly supersymmetric flow equations, which will be presented in section 3, in a simpler context without the complications of fermions.

Nonetheless, even this simplified setting will reveal that new ingredients are required in order to write such flow equations for brane actions in three spacetime dimensions. Unlike the known examples in $d = 2$, where a stress tensor deformation of a free scalar yields the Nambu-Goto Lagrangian [3], and in $d = 4$, where a flow equation deforms the Maxwell Lagrangian into the Born-Infeld theory [62], we will find that flow equations in $d = 3$ require the introduction of a new Lorentz scalar constructed from the stress tensor. This new invariant is the three-dimensional version of the root-$T\overline{T}$ operator [34].

### 2.1 Maxwell-Goldstone multiplet

We begin with the bosonic part of the spacetime action for the supersymmetric $3d$ Born-Infeld theory, which reads

$$S_B = \kappa^2 \int d^3x \left( 1 - \sqrt{1 + \frac{1}{2\kappa^2} f^{ab} f_{ab}} \right). \tag{7}$$

In this section, we follow the same conventions as in Section 4, which are laid out in Appendix B. In particular, spacetime indices are denoted with lowercase Latin letters $a, b$, etc. which are raised and lowered with the flat Minkowski metric $\eta_{ab} = \text{diag}(-1, 1, 1)$. We denote spinorial indices with Greek letters $\alpha, \beta$, etc. which are raised and lowered with $\varepsilon_{\alpha\beta}$ where $\varepsilon_{12} = -1 = -\varepsilon_{21}$.

It will first be convenient to develop some general results for an arbitrary theory of an Abelian field strength $f_{ab}$ in three spacetime dimensions before specializing to the Lagrangian (7). Abelian gauge theories in $d = 3$ are considerably simpler than their counterparts in $d = 4$ because of the smaller number of Lorentz scalars that can be constructed from the field strength. We recall that, in $d = 4$, one can construct two independent Lorentz invariants from the field strength $f_{ab}$, namely

$$-\frac{1}{4} f_{ab} f^{ab} = \frac{1}{2} \left( |\vec{E}|^2 - |\vec{B}|^2 \right), \qquad -\frac{1}{4} f_{ab} \widetilde{f}^{ab} = \vec{E} \cdot \vec{B} \qquad (d = 4), \tag{8}$$

where $\widetilde{f}_{ab} = \frac{1}{2} \epsilon^{abcd} f_{cd}$ is the Hodge dual of $f_{ab}$. A general Lagrangian constructed from the field strength can therefore be written as a function of these two Lorentz scalars. However, in $d = 3$, the analogue of $\widetilde{f}_{ab}$ is a scalar field, and one cannot construct a second independent invariant from the field strength and its dual as in four dimensions. Therefore, a general Lagrangian built from $f_{ab}$ in three dimensions is a function of one variable. We will call this variable $\mathfrak{t}$, not to be confused with the time coordinate $t$, since the combination $\mathfrak{t}$ will be

proportional to the lowest component of the superfield $T$ considered[5] in Sections 3 and 4:

$$\mathcal{L}(f_{ab}) = \mathcal{L}(\mathfrak{t}), \qquad \mathfrak{t} = -\frac{1}{4} f_{ab} f^{ab} \qquad (d = 3). \tag{9}$$

The Hilbert stress tensor $T_{ab}$ associated with such a general Lagrangian $\mathcal{L}(\mathfrak{t})$ is

$$\begin{aligned} T_{ab} &= -2 \frac{\partial \mathcal{L}}{\partial g^{ab}} + g_{ab} \mathcal{L} \\ &= \frac{\partial \mathcal{L}}{\partial \mathfrak{t}} f_a{}^c f_{bc} + g_{ab} \mathcal{L}, \end{aligned} \tag{10}$$

where we take $g_{ab} = \eta_{ab}$ to be the Minkowski metric.

We will need to compute various scalar contractions of the stress-energy tensor in order to construct the operator which drives our flow. Using (10), one finds

$$\begin{aligned} T^{ab} T_{ab} &= 3 \mathcal{L}^2 - 8 \mathcal{L} \mathfrak{t} \frac{\partial \mathcal{L}}{\partial \mathfrak{t}} + 8 \left( \frac{\partial \mathcal{L}}{\partial \mathfrak{t}} \right)^2 \mathfrak{t}^2, \\ T^a{}_a &= 3 \mathcal{L} - 4 \mathfrak{t} \frac{\partial \mathcal{L}}{\partial \mathfrak{t}}. \end{aligned} \tag{11}$$

We have simplified the first expression using the identity

$$f^{ac} f_c{}^b f_a{}^d f_{db} = \frac{1}{2} \left( f_{ab} f^{ab} \right)^2, \tag{12}$$

which follows from the trace relations for a $3 \times 3$ matrix (see [33] or [49] for the computation of the two scalars (11) for a field strength in $d$ spacetime dimensions).

One can now see that there is a special combination of these two scalars:

$$T^{ab} T_{ab} - \frac{1}{3} \left( T^a{}_a \right)^2 = \frac{8}{3} \mathfrak{t}^2 \left( \frac{\partial \mathcal{L}}{\partial \mathfrak{t}} \right)^2. \tag{13}$$

This is an especially nice expression since it depends only on $\mathfrak{t}$ and the derivative $\frac{\partial \mathcal{L}}{\partial \mathfrak{t}}$, but not on the bare Lagrangian $\mathcal{L}$ without any derivatives. Furthermore, since the combination is a perfect square, it is sensible to take its square root so long as we are careful about our choice of branch. To do this, we now lay out our assumptions about the signs of quantities appearing in (13). Recall that

$$\mathfrak{t} = \frac{1}{2} \left( |\vec{E}|^2 - |\vec{B}|^2 \right), \tag{14}$$

and that the Maxwell Lagrangian in our conventions is $\mathcal{L}_{\text{Maxwell}} = \mathfrak{t}$, so that

$$\frac{\partial \mathcal{L}_{\text{Maxwell}}}{\partial \mathfrak{t}} = 1. \tag{15}$$

In particular, for the Maxwell theory we have $\frac{\partial \mathcal{L}}{\partial \mathfrak{t}} > 0$. We will assume that this inequality is true for all of the theories considered in this work. To fix the sign of $\mathfrak{t}$, we further assume that

$$|\vec{E}|^2 < |\vec{B}|^2 \implies \mathfrak{t} < 0. \tag{16}$$

This can be justified, for instance, by focusing on the dynamics of small fluctuations around a classical background with a fixed large magnetic field. In some sense, this is the more natural choice from the perspective of the Born-Infeld theory, where the electric field must still satisfy

---

[5]See, for instance, equations (85) and (170), although note that the conventions for this superfield differ by a factor of $\frac{2}{\kappa^2}$ between these sections.

$|\vec{E}|^2 < \kappa^2$ so that the argument of the square root remains positive. However, the strength of the magnetic field in the Born-Infeld theory is allowed to grow arbitrarily large without affecting the reality of the Lagrangian.

Under the assumptions that $\mathfrak{t} < 0$ and $\frac{\partial \mathcal{L}}{\partial \mathfrak{t}} > 0$, we are free to take the square root of the combination (13) and define

$$
\begin{aligned}
R &= \sqrt{\frac{3}{8}\left(T^{ab}T_{ab} - \frac{1}{3}\left(T^a_{\ a}\right)^2\right)} \\
&= \sqrt{\mathfrak{t}^2\left(\frac{\partial \mathcal{L}}{\partial \mathfrak{t}}\right)^2} \\
&= -\mathfrak{t}\frac{\partial \mathcal{L}}{\partial \mathfrak{t}}\,.
\end{aligned}
\tag{17}
$$

If we had instead assumed $\mathfrak{t} > 0$ and $\frac{\partial \mathcal{L}}{\partial \mathfrak{t}} > 0$, our definition of $R$ would have differed by an overall minus sign, which changes some of the numerical coefficients in the flow equations which follow. One could also have defined a piecewise flow equation, with one choice of coefficients for $\mathfrak{t} < 0$ and another choice for $\mathfrak{t} > 0$, in order to deform the entire phase space of the theory.

*Flow equation*

We are now ready to construct our deforming operator. Consider the flow equation

$$
\begin{aligned}
\frac{\partial \mathcal{L}}{\partial \lambda} &= O_\lambda\,, \\
O_\lambda &= \frac{1}{6}T^{ab}T_{ab} - \frac{1}{9}\left(T^a_{\ a}\right)^2 + \frac{1}{9}\left(T^a_{\ a}\right)R\,,
\end{aligned}
\tag{18}
$$

with $R$ as in (17). Note that we will later use calligraphic symbols like $\mathcal{O}_{\kappa^2}$ for superfield operators, whereas the non-calligraphic symbols like $O_\lambda$ denotes bosonic operators. Here $\lambda$ is an irrelevant coupling constant with length dimension 3.

In this paper, we will always assume that $\lambda$ is positive. This is sometimes referred to as the "good sign" of the deformation parameter because, in the context of $T\overline{T}$ deformations in two spacetime dimensions where the operator is well-defined quantum mechanically, the finite-volume spectrum of a deformed CFT remains real for some range of positive $\lambda$, whereas for negative $\lambda$ all but finitely many of the energy levels become complex.[6] There have been various proposed interpretations for these complex energies, including holographic pictures such as a finite cutoff in AdS$_3$ [64] or a change in spacetime signature [65]. The negative sign of the deformation parameter has also been related to de Sitter constructions [66–68]. However, in the present work we will consider only classical good-sign flows, in part because it is not known whether one can define a local $T\overline{T}$-like operator from the stress tensor in $d > 2$ dimensions (see [69] for a discussion of this point).

Using the explicit expressions (11) for $T^{ab}T_{ab}$, $T^a_{\ a}$ and (17) for $R$, the flow equation (18) can be written as

$$
\frac{\partial \mathcal{L}}{\partial \lambda} = \mathcal{L}\mathfrak{t}\frac{\partial \mathcal{L}}{\partial \mathfrak{t}} - \frac{1}{2}\mathcal{L}^2\,.
\tag{19}
$$

---

[6]In some cases one can cure these complex energies by performing sequential flows, for instance deforming a collection of theories by negative $\lambda$ and then deforming their tensor product by a positive value of $\lambda$ which is sufficiently large [63].

It is also sometimes convenient to change variables as $\lambda = \frac{1}{\kappa^2}$, where $\kappa^2$ has mass dimension 3, and write the flow equation as

$$\kappa^4 \frac{\partial \mathcal{L}}{\partial \kappa^2} = O_{\kappa^2},$$
$$O_{\kappa^2} = -\frac{1}{6} T^{ab} T_{ab} + \frac{1}{9} \left( T^a_{\ a} \right)^2 - \frac{1}{9} \left( T^a_{\ a} \right) R. \tag{20}$$

Because of the change of variables, the operator $O_{\kappa^2} = -O_\lambda$ which drives the flow in equation (20) differs by a sign from the corresponding operator in equation (18).

We now return to the Born-Infeld theory (7). In terms of the variable $\mathfrak{t}$, this Lagrangian can be written as

$$\mathcal{L}_B(\kappa^2, \mathfrak{t}) = \kappa^2 \left( 1 - \sqrt{1 - \frac{2}{\kappa^2} \mathfrak{t}} \right), \tag{21}$$

or using the parameter $\lambda = \frac{1}{\kappa^2}$,

$$\mathcal{L}_B(\lambda, \mathfrak{t}) = \frac{1}{\lambda} \left( 1 - \sqrt{1 - 2\lambda \mathfrak{t}} \right). \tag{22}$$

One can verify by direct computation that the Lagrangian (22) satisfies the differential equation (19), or equivalently that $\mathcal{L}_B(\kappa^2, \mathfrak{t})$ of (21) solves the equation (20).

*Modified trace flow equation*

The free Maxwell theory in three spacetime dimensions is not a conformal field theory. Therefore, deformations of the $3d$ Maxwell Lagrangian – such as the stress tensor flow which produces the Born-Infeld theory that was presented in the preceding subsection – do not exhibit the usual properties for deformations of CFTs which are often used when studying perturbations of this kind.

An especially useful example of such a property is the so-called trace flow equation, which plays a role in many places in the $T\overline{T}$ literature, including in cutoff AdS$_3$ holography [70–75] and in $4d$ $T\overline{T}$-like flows [48]. We now review and extend this result. Let $\mathcal{L}_0$ be a Lagrangian which describes a classically conformal field theory in $d$ spacetime dimensions (in particular, the trace of the Hilbert stress tensor associated with $\mathcal{L}_0$ vanishes). Suppose that we deform $\mathcal{L}_0$ according to the flow equation

$$\frac{\partial \mathcal{L}_\lambda}{\partial \lambda} = O(T_{ab}(\lambda)), \tag{23}$$

with the initial condition $\mathcal{L}_\lambda \to \mathcal{L}_0$ as $\lambda \to 0$, and where $O$ is an arbitrary scalar function of the stress tensor $T_{ab}(\lambda)$ associated with $\mathcal{L}_\lambda$. We assume that the deformation parameter $\lambda$ has length dimension $\Delta$. Because $\mathcal{L}_0$ is conformally invariant, there is only a single energy scale $\Lambda = \lambda^{-1/\Delta}$ in the deformed theory $\mathcal{L}_\lambda$. The response of $\mathcal{L}_\lambda$ to an infinitesimal scale transformation is determined by the trace of the stress tensor as

$$\Lambda \frac{dS}{d\Lambda} = \int d^d x \, T^a_{\ a}(\lambda). \tag{24}$$

Comparing the expression (24) to (23), one concludes that

$$T^a_{\ a}(\lambda) = -\Delta \lambda O(T_{ab}(\lambda)) + \text{total derivative}, \tag{25}$$

where we have written "+ total derivative" because we are equating two spacetime integrals and therefore cannot exclude the possibility that the integrands may differ by total derivative

terms. We will omit such possible terms in the remainder of this subsection, for simplicity, although we will see later around equation (64) that they can be important in some contexts. This equation can also be written as

$$\lambda \frac{\partial \mathcal{L}_\lambda}{\partial \lambda} = -\frac{1}{\Delta} T^a{}_a \,. \tag{26}$$

Therefore, the trace of the stress tensor in the deformed theory $\mathcal{L}_\lambda$ is determined in terms of the deforming operator $O(T_{ab}(\lambda))$. The condition (25) applies, for instance, to the $T\overline{T}$ deformation of a free scalar in two spacetime dimensions, or to the analogous four-dimensional $T^2$ deformation of the free $4d$ Maxwell Lagrangian.

Because the $3d$ Maxwell Lagrangian is not a CFT, the trace flow equation (25) does not hold for the $3d$ Born-Infeld Lagrangian (22), which we have obtained as such a stress tensor flow with $\Delta = d = 3$ and where the operator $O(T_{ab})$ is given by

$$O(T_{ab}) = O_\lambda \equiv \frac{1}{6} T^{ab} T_{ab} - \frac{1}{9} \left( T^a{}_a \right)^2 + \frac{1}{9} \left( T^a{}_a \right) R, \tag{27}$$

as in equation (18). However, one might wonder whether the Born-Infeld theory satisfies a modified version of such a trace flow equation. Besides the trace $T^a{}_a$, another natural Lorentz invariant built from $T_{ab}$ which has the same dimension is the operator

$$R = \sqrt{\frac{3}{8} \left( T^{ab} T_{ab} - \frac{1}{3} \left( T^a{}_a \right)^2 \right)}, \tag{28}$$

which we used to construct the flow. In fact, the Born-Infeld Lagrangian $\mathcal{L}_B$ satisfies

$$\lambda \frac{\partial \mathcal{L}_B}{\partial \lambda} = -\frac{1}{3} \left( T^a{}_a(\lambda) - R(\lambda) \right), \tag{29}$$

which takes a similar form as equation (26) for a conformal seed theory except with an "effective trace" of $T^a{}_a - R$ rather than the usual trace $T^a{}_a$. Equivalently, one finds that the trace of the stress tensor associated with $\mathcal{L}_B$ obeys

$$T^a{}_a - R = -3\lambda O_\lambda \left( T_{ab}(\lambda) \right), \tag{30}$$

for the function $O_\lambda$ in (27). Again, comparing to the ordinary trace flow equation (25) which holds for a deformed CFT, we find that the $3d$ Born-Infeld theory obeys a modified trace flow equation where the role of the trace is played by the difference between the trace $T^a{}_a$ and the root-$T\overline{T}$-like operator $R$.

*Relevant flow equation*

In [48], it was pointed out that one version of the four-dimensional Born-Infeld Lagrangian also satisfies a flow equation driven by a *relevant* operator constructed from the stress tensor, as opposed to the usual irrelevant $T\overline{T}$-like combination which deforms the $4d$ Maxwell theory to Born-Infeld. This result relied upon the fact that one can construct a "subtracted" version of the theory by adding an appropriate $\lambda$-dependent constant to the Lagrangian which causes the $T\overline{T}$-like operator, which usually drives the irrelevant flow, to become a constant. It is natural to ask whether any analogue of these results also holds for the three-dimensional Born-Infeld theory.

We will see that the answer is yes. We first define the "subtracted" $3d$ Born-Infeld Lagrangian as

$$\widetilde{\mathcal{L}}_B = \frac{\alpha}{\lambda} \sqrt{1 - 2\lambda \mathfrak{t}}, \tag{31}$$

where $\alpha$ is a dimensionless constant. We refer to this as a "subtracted" theory because the usual Born-Infeld theory (22) is a sum of two terms, the constant term $\frac{1}{\lambda}$ and the square root term, so one can simply subtract off the constant $\frac{1}{\lambda}$ term (and rescale by an overall factor) to arrive at (31). We note that the equations of motion for this model are not affected by the constant term so the dynamics of $\mathcal{L}_B$ are identical to those of $\widetilde{\mathcal{L}}_B$ (so long as we do not couple the theory to dynamical gravity, in which case the $\frac{1}{\lambda}$ term plays the role of a cosmological constant).

One can compute the stress tensor $\widetilde{T}_{ab}$ associated with the subtracted theory $\widetilde{\mathcal{L}}_B$, and then assemble the combination $O_\lambda(\widetilde{T}_{ab})$, which turns out to be a constant:

$$
\begin{aligned}
O_\lambda(\widetilde{T}_{ab}) &= \frac{1}{6}\widetilde{T}^{ab}\widetilde{T}_{ab} - \frac{1}{9}\left(\widetilde{T}^a{}_a\right)^2 + \frac{1}{9}\left(\widetilde{T}^a{}_a\right)R \\
&= -\frac{\alpha^2}{2\lambda^2}\,.
\end{aligned}
\tag{32}
$$

If we assume that $\lambda$ is positive, we can therefore write

$$
\lambda = \frac{|\alpha|}{\sqrt{2\left|O_\lambda(\widetilde{T}_{ab})\right|}}\,.
\tag{33}
$$

This relation allows us to write a different flow equation for the Lagrangian $\widetilde{\mathcal{L}}_B$. It is most convenient to express this flow in terms of the variable $\kappa^2$ rather than $\lambda = \frac{1}{\kappa^2}$. By first substituting the solution (33) into the modified trace flow equation (30), then using the resulting expression for $T^a{}_a - R$ in equation (29), writing the equation in terms of the stress tensor $\widetilde{T}_{ab}$ for $\widetilde{\mathcal{L}}_B$, and finally changing variables from $\lambda$ to $\kappa^2$, one finds

$$
\frac{\partial \widetilde{\mathcal{L}}_B}{\partial \kappa^2} = \frac{|\alpha|\left(\widetilde{T}^a{}_a - \widetilde{R}\right)}{\sqrt{\left|3\widetilde{T}^{ab}\widetilde{T}_{ab} - 2\left(\widetilde{T}^a{}_a\right)^2 + 2\left(\widetilde{T}^a{}_a\right)\widetilde{R}\right|}}\,,
\tag{34}
$$

where $\widetilde{R}$ is the root-$T\overline{T}$-like combination constructed from $\widetilde{T}_{ab}$. We have therefore written a flow equation for the subtracted Lagrangian $\widetilde{\mathcal{L}}_B$ driven by a relevant operator constructed from the stress tensor.

The non-trivial feature of this result is that the right side of (34) depends only on $\widetilde{T}_{ab}$ and not on the flow parameter $\kappa^2$. One can always trivially rewrite a flow equation involving an irrelevant deformation parameter $\lambda = \frac{1}{\kappa^2}$ by changing variables to $\kappa^2$. However, such a rewriting will generally produce an expression for $\frac{\partial \mathcal{L}}{\partial \kappa^2}$ which is a relevant combination of both $\kappa^2$ and the original deforming operator. In contrast, the flow (34) is driven by an operator built solely from the stress tensor itself, which is a special feature of the Lagrangian $\widetilde{\mathcal{L}}_B$ that is possible because the combination $f(\widetilde{T}_{ab})$ is a constant.

## 2.2 Tensor multiplet

As explored in [30], one can also interpret the three-dimensional $\mathcal{N} = 1$ tensor multiplet as the Goldstone for a spontaneously broken second supersymmetry. In this section, we will study stress tensor flows for the bosonic truncation of this multiplet.

The bosonic field content of the tensor-Goldstone multiplet is a scalar field $\phi$ and an auxiliary field $H$ which can be assembled into a bispinor

$$
\hat{f}_{\alpha\beta} = \partial_{\alpha\beta}\phi + i\varepsilon_{\alpha\beta}H\,,
\tag{35}
$$

where we remind the reader that in this section we use the notation of Appendix B for spinorial indices $\alpha, \beta$.

One could have anticipated that this multiplet, for which the physical bosonic degree of freedom is the scalar field $\phi$, might have provided an equivalent description of the Maxwell-Goldstone multiplet from Hodge duality. Indeed, in three spacetime dimensions a 2-form field strength is equivalent to a scalar field through the duality relation

$$d\phi = \star f \,, \tag{36}$$

where $f = f_{ab} \, dx^a \wedge dx^b$, $d$ is the exterior derivative, and $\star$ is the Hodge star.

The undeformed Lagrangian for the bosonic truncation of this multiplet, which is the large-$\kappa^2$ or small-$\lambda$ limit of the solution to the flow equation which we develop shortly, is proportional to the combination

$$\begin{aligned}
\hat{f}^{\alpha\beta}\hat{f}_{\alpha\beta} &= \partial_{\alpha\beta}\phi\,\partial^{\alpha\beta}\phi + 2H^2 \\
&= -2\partial^a\phi\,\partial_a\phi + 2H^2 \,.
\end{aligned} \tag{37}$$

Note that, for any Lagrangian $\mathcal{L}$ that depends on the field $H$ only through the combination $H^2$ (and which has no kinetic term for $H$), the equation of motion for $H$ is

$$H\frac{\partial\mathcal{L}}{\partial H^2} = 0 \,, \tag{38}$$

which admits the solution $H = 0$. Therefore, if we are only interested in on-shell flows where the equation of motion for $H$ is imposed, then it suffices to eliminate the field $H$ from the Lagrangian and restrict attention to the physical degree of freedom $\phi$. A common issue when studying stress tensor flows for theories which include auxiliary fields is that it is often possible to engineer a flow which reproduces a given Lagrangian on-shell, but not off-shell. We will encounter precisely this issue in the analysis of this section and the same behavior arises in the superspace flow of Section 3.

With a particular choice of normalization, the bosonic truncation of the Lagrangian for the tensor-Goldstone multiplet can be written as

$$\mathcal{L}_{\mathrm{TG}} = \frac{1}{\lambda}\left(1 - \sqrt{1 + 2\lambda\left(\partial^a\phi\,\partial_a\phi - H^2\right)}\right) = \kappa^2\left(1 - \sqrt{1 + \frac{2}{\kappa^2}\left(\partial^a\phi\,\partial_a\phi - H^2\right)}\right), \tag{39}$$

in terms of either the variable $\lambda$ or $\kappa^2 = \frac{1}{\lambda}$. When the auxiliary field $H$ is set to zero using its equation of motion, this reduces to

$$\mathcal{L}_{\mathrm{TG}}\big|_{H=0} = \frac{1}{\lambda}\left(1 - \sqrt{1 + 2\lambda\partial^a\phi\,\partial_a\phi}\right) = \kappa^2\left(1 - \sqrt{1 + \frac{2}{\kappa^2}\partial^a\phi\,\partial_a\phi}\right), \tag{40}$$

which is the scalar sector of the Dirac-Born-Infeld action. From this perspective, $\phi$ can be thought of as a scalar field that describes the transverse fluctuations to a D2-brane, and the parameter $\kappa^2$ controls the tension of the brane.

### *Flow equation*

For the moment, we will consider off-shell flows where we do not eliminate the auxiliary field $H$ using its equation of motion. We first compute the stress tensor for a general Lagrangian $\mathcal{L}(x_1, x_2)$ which depends on the two Lorentz-invariant combinations

$$x_1 = \partial^a\phi\,\partial_a\phi \,, \qquad x_2 = H^2 \,. \tag{41}$$

As $H$ does not couple to the metric, the Hilbert stress tensor is simply

$$T_{ab} = \eta_{ab}\mathcal{L} - 2\frac{\partial\mathcal{L}}{\partial x_1}\partial_a\phi\,\partial_b\phi \,. \tag{42}$$

A direct computation of the contractions of $T_{ab}$ needed to construct our flow gives

$$T^{ab}T_{ab} = 3\mathcal{L}^2 - 4\mathcal{L}x_1\frac{\partial\mathcal{L}}{\partial x_1} + 4x_1^2\left(\frac{\partial\mathcal{L}}{\partial x_1}\right)^2,$$

$$T^a{}_a = 3\mathcal{L} - 2x_1\frac{\partial\mathcal{L}}{\partial x_1}. \tag{43}$$

As in the gauge theory analysis, there is a special combination of these contractions:

$$T^{ab}T_{ab} - \frac{1}{3}\left(T^a{}_a\right)^2 = \frac{8}{3}x_1^2\left(\frac{\partial\mathcal{L}}{\partial x_1}\right)^2. \tag{44}$$

Conveniently, this combination (44) is a perfect square which does not depend directly on $\mathcal{L}$ but only on its derivative with respect to $x_1$.

In order to consistently take square roots, we will again need to make certain assumptions about the signs of quantities in the scalar context. The free Lagrangian is given by $\mathcal{L} = -x_1$, where

$$x_1 = \partial^a\phi\,\partial_a\phi = -(\partial_t\phi)^2 + (\partial_x\phi)^2. \tag{45}$$

We will assume that $x_1 < 0$ and $\frac{\partial\mathcal{L}}{\partial x_1} < 0$. Note that this is, in some sense, the opposite convention as that chosen in section 2.1, where we assumed that $\left|\vec{E}\right|^2 < \left|\vec{B}\right|^2$. Here we are instead supposing that $(\partial_t\phi)^2 > (\partial_x\phi)^2$. We will see that this sign choice is convenient because it will imply that the scalar theory satisfies the same flow equation, with the same relative coefficients, as the Born-Infeld Lagrangian; with the opposite sign convention, the sign of one of the terms would be reversed.

With this choice of sign, we may define the root-$T\overline{T}$-like operator $R$ via the square root of (44),

$$R = \sqrt{\frac{3}{8}\left(T^{ab}T_{ab} - \frac{1}{3}\left(T^a{}_a\right)^2\right)} = x_1\frac{\partial L}{\partial x_1}, \tag{46}$$

where we choose the positive root because $x_1\frac{\partial L}{\partial x_1} > 0$ in these conventions.

It is worth pointing out that, with either choice of sign, the operator $R$ is an on-shell total derivative. The equation of motion associated with any Lagrangian $\mathcal{L}(x_1)$ is

$$0 = \partial_a\left(\frac{\partial\mathcal{L}}{\partial(\partial_a\phi)}\right) = 2\partial^a\left(\partial_a\phi\frac{\partial\mathcal{L}}{\partial x_1}\right). \tag{47}$$

On the other hand,

$$R = x_1\frac{\partial\mathcal{L}}{\partial x_1} = \partial^a\phi\,\partial_a\phi\frac{\partial\mathcal{L}}{\partial x_1} = \partial^a\left(\phi\,\partial_a\phi\frac{\partial\mathcal{L}}{\partial x_1}\right) - \phi\,\partial^a\left(\partial_a\phi\frac{\partial\mathcal{L}}{\partial x_1}\right). \tag{48}$$

Comparing the final expression of (48) to that of (47), we see that the second term of $R$ vanishes on-shell and thus $R$ is a total derivative up to equations of motion as claimed.

We now propose the flow equation

$$\frac{\partial\mathcal{L}}{\partial\lambda} = \frac{1}{6}T^{ab}T_{ab} - \frac{1}{9}\left(T^a{}_a\right)^2 + \frac{1}{9}T^a{}_a \cdot R$$

$$= \mathcal{L}x_1\frac{\partial\mathcal{L}}{\partial x_1} - \frac{1}{2}\mathcal{L}^2, \tag{49}$$

which takes the same form as the differential equation (18) which deformed the free Maxwell theory to the Born-Infeld theory. The solution to the differential equation (49) with initial condition

$$\mathcal{L}_0 = -x_1 + x_2 = -\partial^a\phi\,\partial_a\phi + H^2, \tag{50}$$

is

$$\mathcal{L}(\lambda) = \frac{2 + 2\lambda x_2 - 2\sqrt{1 + \lambda x_1(2 + \lambda x_2)}}{\lambda(2 + \lambda x_2)}. \tag{51}$$

This solution does not match the bosonic sector of the tensor multiplet (39) off-shell. However, if we put the auxiliary on-shell by setting $H = 0$, (51) reduces to

$$\mathcal{L}(\lambda)\big|_{x_2=0} = \frac{1}{\lambda}\left(1 - \sqrt{1 + 2\lambda\partial^a\phi\partial_a\phi}\right), \tag{52}$$

where we have replaced $x_1 = \partial^a\phi\partial_a\phi$. This *does* match the expression (40) for the tensor-Goldstone Lagrangian when the auxiliary field is set to zero. Therefore the differential equation (49) can be thought of as an on-shell flow which produces the tensor-Goldstone Lagrangian from the initial condition $\mathcal{L}_0 = -\partial^a\phi\partial_a\phi + H^2$, albeit only when the auxiliary field is set to zero.

In a sense, the reason that this deformation only yields the desired solution on-shell is because the two terms in $\mathcal{L}_0$ have different metric dependence: the scalar kinetic term $x_1 = g^{ab}\partial_a\phi\partial_b\phi$ couples explicitly to the metric, whereas $x_2 = H^2$ is independent of the metric and therefore contributes to the Hilbert stress tensor only through the variation of the measure factor $\sqrt{-g}$. It is possible, though somewhat artificial, to write a Lagrangian which is equivalent to $\mathcal{L}_0$ but where the two terms couple to the metric in a more symmetrical way by introducing an auxiliary vector field $v^a$; this procedure is discussed in Appendix C and leads to an off-shell flow that yields the tensor-Goldstone Lagrangian.

Let us also make a few comments about dimensional reduction. If we compactify one of the spatial directions of our three-dimensional spacetime on a circle, and reduce the operator driving the flow (49) to two dimensions, the resulting operator is *not* the same as the ordinary two-dimensional $T\overline{T}$ operator. This is clear both because the numerical coefficients multiplying the first two terms will differ from those of the $2d$ operator (1), and because the third term will survive, which is not present in $\mathcal{O}^{(2d)}_{T\overline{T}}$. In particular, there is no reason to expect that this dimensionally-reduced $3d$ operator will share any of the desirable properties of the usual $2d$ $T\overline{T}$ operator, such as preserving integrability.

This observation is in agreement with the conclusions of [76], which studied the dimensional reduction of the $3d$ membrane theory to $2d$, and found that the S-matrix of the resulting two-dimensional theory is *not* integrable. However, the same authors found that performing a conventional $2d$ $T\overline{T}$ deformation of the free limit of this dimensionally reduced theory *does* yield an integrable S-matrix, as it must. Because our flow equation (49) yields the $3d$ membrane theory, it is therefore expected that its dimensional reduction differs from the $2d$ $T\overline{T}$ deformation, since this dimensional reduction cannot preserve integrability in light of the results of [76].

*Modified trace flow equation*

For simplicity, in this section we restrict to on-shell flows where the auxiliary field $H$ is set to zero. We focus on the flow for the Lagrangian

$$\mathcal{L} = \frac{1}{\lambda}\left(1 - \sqrt{1 + 2\lambda x_1}\right), \tag{53}$$

where $x_1 = \partial^a\phi\partial_a\phi$, which we have seen satisfies the flow equation

$$\frac{\partial L}{\partial \lambda} = O_\lambda = \frac{1}{6}T^{ab}T_{ab} - \frac{1}{9}\left(T^a_{\ a}\right)^2 + \frac{1}{9}T^a_{\ a}R, \tag{54}$$

where $R = \sqrt{\frac{3}{8}\left(T^{ab}T_{ab} - \frac{1}{3}\left(T^a_{\ a}\right)^2\right)} = x_1\frac{\partial\mathcal{L}}{\partial x_1}$.

The undeformed limit of (53) is the theory of a free scalar in three spacetime dimensions, which is not scale invariant in the sense of having a stress tensor with vanishing trace. For $\mathcal{L}_0 = -\partial^a \phi \partial_a \phi$, one finds

$$T^a_{\ a} = -\partial^a \phi \partial_a \phi \neq 0 \,. \tag{55}$$

Although one can perform an improvement transformation to eliminate the trace (55) for the free scalar theory (which is a consequence of the fact that $\partial^a \phi \partial_a \phi$ is an on-shell to- tal derivative), the standard Hilbert stress tensor without any improvement term is non- vanishing. As a result, the deformed theory (53) cannot possibly satisfy the trace flow equation $T^a_{\ a}(\lambda) = -3\lambda f(T_{ab}(\lambda))$ for a deformed conformal field theory, where $f(T_{ab}(\lambda))$ is defined by equation (54), since this requires $T^a_{\ a} = 0$ when $\lambda = 0$.

However, as in the analysis of Section 2.1, one might ask whether the deformed Lagrangian (53) satisfies a modified version of the usual trace flow equation as the flow parameter $\lambda$ is varied. Again we find that the answer is yes; one finds that this Lagrangian satisfies the relations

$$\lambda \frac{\partial \mathcal{L}}{\partial \lambda} = -\frac{1}{3} \left( T^a_{\ a} - R \right) , \tag{56}$$

or equivalently

$$\lambda O_\lambda (T_{ab}(\lambda)) = -\frac{1}{3} \left( T^a_{\ a} - R \right) , \tag{57}$$

where again $O_\lambda (T_{ab}(\lambda)) = \frac{1}{6} T^{ab} T_{ab} - \frac{1}{9} \left( T^a_{\ a} \right)^2 + \frac{1}{9} T^a_{\ a} R$ is the deforming operator. Thus we see that this deformation of the three dimensional free scalar satisfies a modified trace flow equation where the role of the trace is played by the combination $T^a_{\ a} - R$.

Equations (56) and (57) hold as exact off-shell relations. However, we have noted before around equation (48) that the operator $R$ is an on-shell total derivative for any Lagrangian $\mathcal{L}(x_1)$. Thus the contribution from the $R$ term drops out of the integrated expressions of these modified trace flow equations. For instance, one has

$$\int d^3 x \, \lambda \frac{\partial \mathcal{L}}{\partial \lambda} = -\frac{1}{3} \int d^3 x \left( T^a_{\ a} - R \right) \simeq -\frac{1}{3} \int d^3 x \, T^a_{\ a} \,, \tag{58}$$

where in the last step we write $\simeq$ to indicate equivalence up to any possible boundary terms which arise from integrating a total spacetime derivative and equations of motion.

This is in accord with the fact that, despite the non-zero trace of the Hilbert stress tensor for the theory of a free scalar field in $d > 2$ dimensions, one can define an improved stress tensor whose trace vanishes. Let us briefly review this simple observation. In any spacetime dimension $d$, the Lagrangian $\mathcal{L} = \partial^a \phi \partial_a \phi$ for a single massless scalar has the Hilbert stress tensor

$$T_{ab} = -2\partial_a \phi \partial_b \phi + g_{\mu\nu} \partial^c \phi \partial_c \phi \,, \tag{59}$$

whose trace is $T^a_{\ a} = (d-2)\partial^c \phi \partial_c \phi$. One can always improve the stress tensor as

$$T_{ab} \longrightarrow T'_{ab} = T_{ab} + (\partial_a \partial_b - \eta_{ab} \partial^c \partial_c) u \,, \tag{60}$$

for any function $u$, without affecting the symmetry or conservation of the stress tensor. In particular, we may choose

$$u = \frac{d-2}{2(d-1)} \phi^2 \,, \tag{61}$$

so that the improved stress tensor is

$$T'_{ab} = -2\partial_a \phi \partial_b \phi + g_{\mu\nu} \partial^c \phi \partial_c \phi + \frac{d-2}{2(d-1)} (\partial_a \partial_b - \eta_{ab} \partial^c \partial_c) \phi^2 \,. \tag{62}$$

The trace of the stress tensor, after dropping all terms proportional to $\partial^c \partial_c \phi$ which vanish on-shell, is then

$$T'^a{}_a = \left(-2 + d + \frac{d-2}{2(d-1)} \cdot 2(1-d)\right) \partial^c \phi \partial_c \phi = 0 \,. \tag{63}$$

Now consider deforming this $d$-dimensional free scalar theory by any function $f\left(T'_{ab}\right)$. Since the improved stress tensor $T'_{ab}$ agrees with the Hilbert stress tensor $T_{ab}$ up to on-shell total derivatives, it can also be used to generate scale transformations inside of spacetime integrals. Further, the trace of the improved stress tensor vanishes at $\lambda = 0$ by construction. Thus by the general arguments presented around equation (25), one has

$$\lambda \int d^3x \, \frac{\partial \mathcal{L}_\lambda}{\partial \lambda} = -\frac{1}{d} \int d^3x \, T'^a{}_a \,. \tag{64}$$

If one were to strip off the integrals in (64), equating the integrands and neglecting total derivatives, one would conclude that $\lambda \frac{\partial \mathcal{L}_\lambda}{\partial \lambda} = -\frac{1}{d} T'^a{}_a$, which seems to contradict the result (56) because it is missing the term proportional to $R$. However we now see that the two equations are consistent inside of a total spacetime integral, because the term $R$ is an on-shell total derivative, as is the difference between $T'^a{}_a$ and $T^a{}_a$.

### Relevant flow equation

For completeness, we now repeat the relevant flow analysis of Section 2.1 for the case of the deformed scalar theory. We will find that a subtracted version of this Lagrangian also satisfies a flow equation driven by a relevant combination of stress tensors, exactly like in the gauge theory case. As in the preceding analysis, for simplicity we will use the auxiliary field equation of motion to set $H = 0$ and focus purely on the scalar $\phi$.

We first define the subtracted version of the deformed scalar Lagrangian as

$$\widetilde{\mathcal{L}} = \frac{\alpha}{\lambda} \sqrt{1 + 2\lambda x_1} \,, \tag{65}$$

where $\alpha$ is a dimensionless constant. Again, the constant term that we have removed does not affect the dynamics of the theory. However, we note that for the $2d$ $T\overline{T}$ deformation of free scalars, the corresponding $\frac{1}{\lambda}$ term can be interpreted as arising from a coupling between the Nambu-Goto string and a constant target-space $B$ field [77]. A related observation is that the $2d$ $T\overline{T}$ deformation can be obtained by considering the uniform light-cone gauge for the string, as discussed in [52, 78–80]. In our case, since we may interpret the scalar $\phi$ as a transverse fluctuation of a D2-brane, it is natural to instead view the $\frac{1}{\lambda}$ term as a coupling to a constant target-space Ramond-Ramond field $C_3$. It would be interesting to explore whether an analogue of the uniform-light cone gauge for a D2-brane yields the deformation considered in this work.

Computing the stress tensor $\widetilde{T}_{ab}$ associated with the subtracted Lagrangian $\widetilde{\mathcal{L}}$, one finds that it satisfies

$$O_\lambda\left(\widetilde{T}_{ab}(\lambda)\right) = \frac{1}{6} \widetilde{T}^{ab} \widetilde{T}_{ab} - \frac{1}{9} \left(\widetilde{T}^a{}_a\right)^2 + \frac{1}{9} \widetilde{T}^a{}_a \widetilde{R} = -\frac{\alpha^2}{2\lambda} \,, \tag{66}$$

where $\widetilde{R}$ is the root-$T\overline{T}$-like operator for the subtracted theory $\widetilde{\mathcal{L}}$. In particular, the deforming operator $O_\lambda\left(\widetilde{T}_{ab}(\lambda)\right)$ of equation (66) is a constant, independent of fields.

By an argument analogous to that of section 2.1, one can then show that the subtracted scalar Lagrangian satisfies

$$\frac{\partial \widetilde{\mathcal{L}}}{\partial \kappa^2} = \frac{|\alpha| \left(\widetilde{T}^a{}_a - \widetilde{R}\right)}{\sqrt{\left|3T^{ab}T_{ab} - 2\left(T^a{}_a\right)^2 + 2T^a{}_a R\right|}} \,, \tag{67}$$

which is again a flow equation driven by a relevant combination of stress tensors.

*General dimension*

To conclude this section, we point out that the possibility of engineering a square-root solution to the flow equation for a single free scalar field is not special to three spacetime dimensions. A straightforward generalization of the above argument applies in any spacetime dimension $d$.

Let $\phi$ be a scalar field in $d$ dimensions and consider a general Lagrangian which depends on the Lorentz invariant $x_1 = \partial^a \phi \partial_a \phi$. The Hilbert stress tensor is identical to the one we considered in $d = 3$, namely

$$T_{ab} = \eta_{ab}\mathcal{L} - 2\frac{\partial \mathcal{L}}{\partial x_1}\partial_a \phi \partial_b \phi \,, \tag{68}$$

and the contractions we need are

$$T^{ab}T_{ab} = d\mathcal{L}^2 - 4\mathcal{L}x_1\frac{\partial \mathcal{L}}{\partial x_1} + 4x_1^2\left(\frac{\partial \mathcal{L}}{\partial x_1}\right)^2 \,,$$
$$T^a_{\ a} = d\mathcal{L} - 2x_1\frac{\partial \mathcal{L}}{\partial x_1} \,. \tag{69}$$

Now the special combination of these invariants is

$$T^{ab}T_{ab} - \frac{1}{d}\left(T^a_{\ a}\right)^2 = \frac{4(d-1)}{d}x_1^2\left(\frac{\partial \mathcal{L}}{\partial x_1}\right)^2 \,. \tag{70}$$

If we again assume $x_1 < 0$ and $\frac{\partial \mathcal{L}}{\partial x_1} < 0$ then we may take the square root to define

$$R = \sqrt{\frac{d}{4(d-1)}\left(T^{ab}T_{ab} - \frac{1}{d}\left(T^a_{\ a}\right)^2\right)} = x_1\frac{\partial \mathcal{L}}{\partial x_1} \,. \tag{71}$$

Next we consider the flow equation

$$\frac{\partial \mathcal{L}}{\partial \lambda} = O^{(d)}_\lambda \,,$$
$$O^{(d)}_\lambda = \frac{1}{2d}T^{ab}T_{ab} - \frac{1}{d^2}\left(T^a_{\ a}\right)^2 + \frac{d-2}{d^2}T^a_{\ a}R, \tag{72}$$

which simplifies to

$$\frac{\partial \mathcal{L}}{\partial \lambda} = -\frac{1}{2}\mathcal{L}^2 + \mathcal{L}x_1\frac{\partial \mathcal{L}}{\partial x_1} \,. \tag{73}$$

The solution to this differential equation with initial condition $\mathcal{L}_0 = -x_1 = -\partial^a \phi \partial_a \phi$ is

$$\mathcal{L}(\lambda) = \frac{1}{\lambda}\left(1 - \sqrt{1 + 2\lambda x_1}\right) \,. \tag{74}$$

As in the three-dimensional case, this $d$-dimensional Lagrangian satisfies a modified trace flow equation

$$\lambda\frac{\partial \mathcal{L}}{\partial \lambda} = -\frac{1}{d}\left(T^a_{\ a} - (d-2)R\right) \,, \tag{75}$$

or equivalently

$$\lambda O^{(d)}_\lambda(T_{ab}(\lambda)) = -\frac{1}{d}\left(T^a_{\ a} - (d-2)R\right) \,. \tag{76}$$

By an argument identical to that in equations (47) and (48), the operator $R$ is an on-shell total derivative in the $d$-dimensional setting, so this modified trace flow equation is consistent with

the fact that the stress tensor for the undeformed theory $\mathcal{L}_0 = -\partial^a \phi \partial_a \phi$ can be improved in such a way to make it traceless.

Finally, one can repeat the analysis of the subtracted version of the scalar Lagrangian in $d$ spacetime dimensions, letting

$$\widetilde{\mathcal{L}} = \frac{\alpha}{\lambda} \sqrt{1 + 2\lambda x_1}, \tag{77}$$

which satisfies

$$O_\lambda^{(d)}\big(\widetilde{T}_{ab}(\lambda)\big) = \frac{1}{2d} T^{ab} T_{ab} - \frac{1}{d^2}\big(T^a_{\ a}\big)^2 + \frac{d-2}{d^2} T^a_{\ a} R = -\frac{\alpha^2}{2\lambda}, \tag{78}$$

As a consequence, when written in terms of the variable $\kappa^2 = \frac{1}{\lambda}$, the subtracted Lagrangian (77) also satisfies a relevant stress tensor flow:

$$\frac{\partial \widetilde{\mathcal{L}}}{\partial \kappa^2} = \frac{|\alpha|\big(\widetilde{T}^a_{\ a} - (d-2)\widetilde{R}\big)}{\sqrt{\left| d T^{ab} T_{ab} - 2\big(T^a_{\ a}\big)^2 + 2(d-2) T^a_{\ a} R \right|}}. \tag{79}$$

## 3 Supersymmetric flows

In this section, we consider the $\mathcal{N} = 1$ supersymmetric extension of the 3d Born-Infeld action and investigate its interpretation as a flow that deforms the free super-Maxwell theory. We begin with the 3d $\mathcal{N} = 1$ Maxwell-Goldstone (MG) multiplet [30]. Recall that in three dimensions, the $\mathcal{N} = 1$ vector multiplet is described by the following superspace action[7]

$$S_{\mathrm{VM}} \sim \int d^3 x \, d^2\theta \, W^2, \quad W^2 := \frac{1}{2} W^\alpha W_\alpha, \tag{80}$$

where the superfield strength $W_\alpha$ is constrained by the Bianchi identity

$$\mathrm{D}^\alpha W_\alpha = 0 \quad \Rightarrow \quad \mathrm{D}^2 W_\alpha = i\partial_\alpha^{\ \beta} W_\beta, \quad \mathrm{D}_\alpha W_\beta = \mathrm{D}_\beta W_\alpha. \tag{81}$$

This Bianchi identity can be solved by expressing $W_\alpha$ in terms of an unconstrained prepotential superfield $\Gamma_\beta$

$$W_\alpha = \frac{1}{2} \mathrm{D}^\beta \mathrm{D}_\alpha \Gamma_\beta. \tag{82}$$

$\Gamma_\beta$ is not uniquely determined and is defined modulo the following gauge transformation

$$\delta \Gamma_\alpha = \mathrm{D}_\alpha K, \tag{83}$$

where $K$ is an arbitrary scalar superfield. This vector supermultiplet will play the role of the Goldstone multiplet associated with the spontaneous supersymmetry breaking $\mathcal{N}=2 \to \mathcal{N}=1$. The dynamics of MG are determined by requiring the surviving supersymmetry be manifest and the broken one realized nonlinearly. The resulting Lagrangian is

$$\mathcal{L}_{\kappa^2} = W^2 f(T), \tag{84}$$

where

$$f(T) = 1 + \frac{T}{1 - T + \sqrt{1 - 2T}}, \quad T = \frac{2}{\kappa^2} \mathrm{D}^2 W^2. \tag{85}$$

---

[7]In this section, we follow the conventions of [30] and *Superspace* [81] – see Appendix A for details.

The dimensionful parameter $\kappa$ ($[\kappa] = 3/2$) corresponds to the VEV of the *superfield F*, which is one of the partners of $W_\alpha$ under the second supersymmetry transformation $\delta^*$ [30]:

$$\delta^*_\epsilon W_\alpha = \epsilon_\alpha F + \cdots, \quad \langle F \rangle \sim \kappa. \tag{86}$$

In other words, $\kappa$ parametrizes by how much we break the second supersymmetry. In particular, at the $\kappa \to \infty$ limit, the theory reduces to the $\mathcal{N} = 1$ free Maxwell theory (80). The dependence of the Lagrangian (84) on $\kappa$ defines a curve in the space of supersymmetric field theories. This curve can be interpreted as a flow. The operator that triggers this flow is defined as

$$\mathcal{O}_{\kappa^2} := \frac{\partial \mathcal{L}_{\kappa^2}}{\partial \kappa^2} = W^2 \frac{\partial f(T)}{\partial \kappa^2} = W^2 f'(T) \frac{\partial T}{\partial \kappa^2} = -\frac{1}{\kappa^2} W^2 T f'(T). \tag{87}$$

In Section 2, it was shown that the $3d$ Born-Infeld theory, which is the bosonic truncation of the MG theory, can be obtained by a flow driven by the (stress tensor)$^2$-type operator. We would like to investigate whether this statement holds true for its supersymmetric extension given by (84). Namely, can the flow operator (87) be written in terms of the supersymmetric extension of the stress tensor?

We will show explicitly that the answer to the above question is positive. The proof is organized in the following way. In section 3.1, we introduce the supercurrent multiplet and give its explicit component field expansion which includes the stress tensor and its supersymmetric partner. Section 3.2 is devoted to the derivation of the supercurrent multiplet superfields for the Maxwell-Goldstone theory and in section 3.3 we explicitly check that the flow operator (87) is indeed a linear combination of (supercurrent)$^2$ terms. We also check that the bosonic truncation of this result matches with section 2.1. As discussed in [30], the role of the Goldstone multiplet can also be played by the three dimensional projection of the tensor multiplet which is dual to the vector multiplet. In section 3.4, we repeat the above calculations for the tensor-Goldstone (TG) description. We confirm that for this case, the flow operator also takes the (supercurrent)$^2$ form and is consistent with section 2.2.

## 3.1 Supercurrent multiplet and stress tensor

For any field theory, one can use the standard formula for the Hilbert stress tensor to find $T_{\underline{ab}}$, as we did in section 2. However, in supersymmetric theories the physical degrees of freedom are organized into supersymmetric multiplets, and therefore the stress tensor $T_{\underline{ab}}$ becomes a member of such a multiplet which we call the supercurrent multiplet.[8]

The stress tensor of a field theory, by definition, couples the theory to gravity at cubic level. Similarly, the supercurrent multiplet of a supersymmetric theory $\mathcal{L}_0[W]$ defines the coupling of the theory with linearized supergravity in the following manner:

$$\mathcal{L}_0[W] \mapsto \mathcal{L}[W, \Psi_{\alpha\beta\gamma}, \varphi] = \mathcal{L}_0[W] + \Psi^{\alpha\beta\gamma} J_{\alpha\beta\gamma}[W] + \varphi J[W] + \mathcal{L}_{SG}^{\text{Linearized}}[\Psi_{\alpha\beta\gamma}, \varphi]. \tag{88}$$

The superfield $\Psi_{\alpha\beta\gamma}$ is a completely symmetric spinor and serves as the linearized prepotential for $3d$, $\mathcal{N} = 1$ conformal supergravity and $\varphi$ is the linearized compensator. The superfields $J_{\alpha\beta\gamma}[W]$ and $J[W]$ define the supercurrent multiplet of the theory $\mathcal{L}_0[W]$ and they are called the supercurrent and supertrace, respectively.

The supergravity superfields $\Psi^{\alpha\beta\gamma}$, $\varphi$ enjoy the following gauge transformations

$$\begin{aligned}
\delta \Psi^{\alpha\beta\gamma} &\sim D^{(\alpha} \xi^{\beta\gamma)}, \\
\delta \varphi &\sim \partial_{\alpha\beta} \xi^{\alpha\beta},
\end{aligned} \tag{89}$$

---

[8]Famous examples of $4d$ supercurrent multiplets are the Ferrara-Zumino multiplet [82], the $\mathcal{R}$-multiplet (see section 7 of [81]) and their generalization the S-multiplet [83,84]. Also see [85] and related constructions in [86]. For $3d$, which is relevant for our paper, the $\mathcal{N} = 2$ supercurrents were discussed in [87] while the $\mathcal{N} = 1$ case is directly related to the $2d$ $\mathcal{N} = (1,1)$ case of [52,53].

where $\xi^{\alpha\beta}$ is an arbitrary, symmetric, superfield parameter for the above linearized superdiffeomorphism.[9]

The gauge invariance of (88) requires the following superspace conservation equation of the supercurrent multiplet — modulo equations of motion —

$$D^\alpha J_{\alpha\beta\gamma}(x,\theta) = 2i\,\partial_{\beta\gamma}J(x,\theta)\,. \qquad (90)$$

This is the supersymmetric extension of the stress tensor conservation equation.

**Improvement terms**   It is important to emphasize that the above conservation equation does not uniquely define the supercurrent and supertrace superfields. In fact there is an infinite family of them. This is understood by the so-called improvement terms. These are the deformations that identically satisfy (90). It is straightforward to check that the supercurrent multiplet $(\tilde{J}_{\alpha\beta\gamma},\tilde{J})$ defined as follows

$$\begin{aligned}
\tilde{J}_{\alpha\beta\gamma} &= J_{\alpha\beta\gamma} + D_{(\alpha}\partial_{\beta\gamma)}\Lambda\,,\\
\tilde{J} &= J - 4i\,D^2\Lambda\,,
\end{aligned} \qquad (91)$$

where $\Lambda$ is an arbitrary superfield also satisfies conservation equation (90). These are the supersymmetric extension of (60).

**Components of supercurrent multiplet**   The $\theta$-expansions for the superfields $J_{\alpha\beta\gamma}(x,\theta)$ and $J(x,\theta)$ are the following

$$\begin{aligned}
J_{\alpha\beta\gamma}(x,\theta) &= s_{\alpha\beta\gamma}(x) + \theta^\rho\left[t_{\rho\alpha\beta\gamma}(x) - \frac{i}{4}C_{\rho(\alpha}\partial_{\beta\gamma)}j(x)\right]\\
&\quad + \theta^2\left[\frac{2i}{3!}\partial_{(\alpha\beta}s_{\gamma)}(x) - \frac{i}{3!}\partial_{(\alpha}{}^\rho s_{\beta\gamma)\rho}(x)\right]\,,\\
J(x,\theta) &= j(x) + \theta^\alpha s_\alpha(x) - \frac{1}{3}\theta^2\Theta(x)\,.
\end{aligned} \qquad (92)$$

Here the component fields $s_{\alpha\beta\gamma}(x)$ and $t_{\alpha\beta\gamma\delta}(x)$ are completely symmetric on their spinorial indices. Moreover, (90) implies that the following fields

$$T_{\alpha\beta\gamma\delta}(x) := t_{\alpha\beta\gamma\delta}(x) - \frac{1}{3}C_{\gamma(\alpha}C_{\beta)\delta}\Theta(x)\,, \qquad (93a)$$

$$S_{\alpha\beta\gamma}(x) := s_{\alpha\beta\gamma}(x) - C_{\gamma(\alpha}s_{\beta)}(x)\,, \qquad (93b)$$

satisfy the conservation equations

$$\partial^{\alpha\beta}S_{\alpha\beta\gamma}(x) = 0\,, \quad \partial^{\rho\alpha}T_{\rho\alpha\beta\gamma}(x) = 0\,. \qquad (94)$$

The field $T_{\alpha\beta\gamma\delta}(x)$ corresponds to the stress tensor and its spinor indices have the following symmetry properties: $T_{\alpha\beta\gamma\delta}(x) = T_{\beta\alpha\gamma\delta}(x) = T_{\gamma\delta\alpha\beta}(x)$. Using (A.28), the conversion between spacetime indices and spinorial indices is:

$$T_{\underline{ab}} := -\frac{1}{4}(\gamma_{\underline{a}})^{\alpha\beta}(\gamma_{\underline{b}})^{\gamma\delta}T_{\alpha\beta\gamma\delta}\,, \quad T_{\alpha\beta\gamma\delta} = T_{\gamma\delta\alpha\beta} = -(\gamma^{\underline{a}})_{\alpha\beta}(\gamma^{\underline{b}})_{\gamma\delta}T_{\underline{ab}}\,. \qquad (95)$$

---

[9]The superfield $\Psi_{\alpha\beta\gamma}$ is the linearized super-vielbein, hence $\nabla_\alpha = \Psi_{\alpha\beta\gamma}\partial^{\beta\gamma} + \cdots$ This determines the engineering dimensions $[\Psi^{\alpha\beta\gamma}] = -1/2$ and via (88) $[J_{\alpha\beta\gamma}] = 5/2$. The conservation equation (90) fixes $[J] = 2$.

Note that $t_{\alpha\beta\gamma\delta}(x)$ is the completely symmetric part of the $T_{\alpha\beta\gamma\delta}(x)$ and it appears only in the supercurrent superfield $J_{\alpha\beta\gamma}$, while $\Theta(x)$ is the trace part of $T_{\underline{ab}}$ and it appears only in the supertrace superfield $J$.

$$t_{\alpha\beta\gamma\delta}(x) = \frac{1}{4!} T_{(\alpha\beta\gamma\delta)}(x) = \frac{1}{4!} D_{(\alpha} J_{\beta\gamma\delta)}(x,\theta)|,$$
$$\Theta(x) = T_{\underline{a}}{}^{\underline{a}} = \frac{1}{2} T_{\alpha\beta}{}^{\alpha\beta} = 3 D^2 J(x,\theta)|. \tag{96}$$

One can also check that the traceless part $\widehat{T}_{\underline{ab}}$ of the stress tensor only depends on $t_{\alpha\beta\gamma\delta}(x)$,

$$\widehat{T}_{\underline{ab}} = T_{\underline{ab}} - \frac{1}{3} \eta_{\underline{ab}} \Theta = -\frac{1}{4} (\gamma_{\underline{a}})^{\alpha\beta} (\gamma_{\underline{b}})^{\gamma\delta} t_{\alpha\beta\gamma\delta}(x), \tag{97}$$

and using (A.15) the square of the traceless part becomes

$$\widehat{T}^{\underline{ab}} \widehat{T}_{\underline{ab}} = T^{\underline{ab}} T_{\underline{ab}} - \frac{1}{3} \Theta^2 = \frac{1}{4} t_{\alpha\beta\gamma\delta} t^{\alpha\beta\gamma\delta}. \tag{98}$$

The operator $S_{\alpha\beta\gamma}(x) = S_{\alpha\gamma\beta}(x)$ is the supersymmetric partner of $T_{\alpha\beta\gamma\delta}(x)$ and it corresponds to Noether's conserved current associated with supersymmetry. Its completely symmetric part $s_{\alpha\beta\gamma}(x)$ is embedded in the supercurrent, while its trace $s_\alpha(x)$ is embedded in the supertrace.

$$s_{\alpha\beta\gamma}(x) = \frac{1}{3!} S_{(\alpha\beta\gamma)}(x) = J_{\alpha\beta\gamma}(x,\theta)|,$$
$$s_\alpha(x) = \frac{1}{3} S_{\alpha\beta}{}^\beta(x) = D_\alpha J(x,\theta)|. \tag{99}$$

## 3.2 Supercurrent and supertrace of 3$d$ Maxwell-Goldstone multiplet

The explicit calculation of the supercurrent multiplet $(J_{\alpha\beta\gamma}, J)$ for a particular theory can be carried out in various ways. The typical method, as suggested by (88), is to couple the theory to supergravity and then take the linearized limit in order to read off the supercurrent $J_{\alpha\beta\gamma}$ and the supertrace $J$. This approach will be followed in section 4. A different methodology is to use the superspace conservation equation (90) as a consistency condition which can be solved in order to determine the supercurrent multiplet. This latter approach will be used in this section for MG theory (84). The calculation has two parts. First note that for $f(T) = 1$ ($\kappa \to \infty$), equation (84) reduces to the free super-Maxwell theory (80). Therefore we start with a warm-up calculation of the supercurrent multiplet corresponding to the free vector supermultiplet in order to build our intuition about the structure of the supercurrent and supertrace. In the second part, we generalize these results to arbitrary $f(T)$ (finite $\kappa \neq 0$).

### 3.2.1 Warm up: Supercurrent multiplet of super-Maxwell

For the free super-Maxwell theory, the supercurrent multiplet $\{J_{\alpha\beta\gamma}, J\}$ must be quadratic in $W_\alpha$.[10] Moreover, due to the engineering dimensions $[J_{\alpha\beta\gamma}] = 5/2$, $[J] = 2$, and $[W_\alpha] = 1$, the supercurrent will include exactly one spinorial derivative and the supertrace will have zero spinorial derivatives. Finally, the index structure of $J_{\alpha\beta\gamma}$ and $J$ suggests the following ansatz[11]

$$J_{\alpha\beta\gamma} = W_{(\alpha} D_\beta W_{\gamma)}, \quad J = W^2. \tag{100}$$

---

[10] The use of bare prepotential superfield $\Gamma_\alpha$ (82) is not allowed due to the invariance of the cubic vertex (88) under the gauge transformation (83).

[11] The corresponding supercurrent of the 4$d$ vector multiplet was derived in [82], see also [88]. Extensions to higher spin gauge theories are found in [89,90].

We will now show that (100) indeed satisfies the conservation equation (90) up to the equations of motion:

$$D^\beta D_\alpha W_\beta = 0 \quad \overset{(81)}{\Rightarrow} \quad \partial_{\alpha\beta} W^\beta = 0 = D^2 W_\alpha. \tag{101}$$

Direct computation yields

$$\begin{aligned} D^\alpha J_{\alpha\beta\gamma} = {} & (D^\alpha W_\alpha)(D_{(\beta} W_{\gamma)}) + (D^\alpha W_\beta)(D_{(\gamma} W_{\alpha)}) + (D^\alpha W_\gamma)(D_{(\alpha} W_{\beta)}) \\ & - W_\alpha D^\alpha D_{(\beta} W_{\gamma)} - W_\beta D^\alpha D_{(\gamma} W_{\alpha)} - W_\gamma D^\alpha D_{(\alpha} W_{\beta)}. \end{aligned} \tag{102}$$

Note that (1.) by using (81), the first term vanishes; (2.) terms two and three cancel; (3.) terms five and six vanish independently on-shell by (101); (4.) the fourth term can be simplified in the following way via (A.21) and (101)

$$D^\alpha J_{\alpha\beta\gamma} = W^\alpha D_\alpha D_{(\beta} W_{\gamma)} = i\, W^\alpha\, \partial_{\alpha(\beta} W_{\gamma)} = i\, \partial_{\alpha(\beta} W^\alpha W_{\gamma)} = 2i\, \partial_{\beta\gamma} W^2. \tag{103}$$

This is the conservation equation (90).

### 3.2.2 Supercurrent multiplet of Maxwell-Goldstone

For the Maxwell-Goldstone theory

$$S_{\mathrm{MG}} = \int d^3x\, d^2\theta\, W^2 f(T), \tag{104}$$

$f(T)$ (85) is no longer a constant and therefore its supercurrent multiplet $\{\mathcal{J}_{\alpha\beta\gamma}, \mathcal{J}\}$ will generalize the super-Maxwell theory results (100) with additional terms depending on the derivatives of $f(T)$, such that in the $f(T) \to 1$ limit we restore the results of section 3.2.1.

$$\begin{aligned} \mathcal{J}_{\alpha\beta\gamma} &\sim f(T)\, J_{\alpha\beta\gamma} + f'(T)\, \mathcal{O}^{(1)}_{\alpha\beta\gamma} + f''(T)\, \mathcal{O}^{(2)}_{\alpha\beta\gamma} + \cdots, \\ \mathcal{J} &\sim f(T)\, J + f'(T)\, \mathcal{O}^{(1)} + f''(T)\, \mathcal{O}^{(2)} + \cdots \end{aligned} \tag{105}$$

This is also reflected in the fact that the equations of motion of MG theory include factors of $f(T)$ and its derivatives.

**EOM** Indeed, by varying the action (104) with respect to the prepotential superfield $\Gamma_\beta$, it is straightforward to find the following equation of motion

$$D^\rho D_\alpha \big[ W_\rho\, g \big] = 0, \tag{106}$$

where the factor $g$ is defined as

$$\begin{aligned} g &:= f(T) + \frac{2}{\kappa^2} D^2 \big[ W^2 f'(T) \big] \\ &= f(T) + T f'(T) + \frac{2}{\kappa^2} W^2 D^2 f'(T) + \frac{2}{\kappa^2} (D^\delta W^2)(D_\delta f'(T)). \end{aligned} \tag{107}$$

Equivalently, using (A.21) the equation of motion can also take the form

$$D^2 \big\{ W_\alpha\, g \big\} = -i\, \partial_\alpha{}^\beta \big\{ W_\beta\, g \big\}. \tag{108}$$

As expected, (106) reduces to (101) when we take $f \to 1$.

**Supercurrents**   The results of section 3.2.1 in combination with the MG equations of motion suggest the following ansatz for the MG supercurrent

$$\mathcal{J}^{(1)}_{\alpha\beta\gamma} = W_{(\alpha}\big(\mathrm{D}_\beta W_{\gamma)}\big)g = W_{(\alpha}\mathrm{D}_\beta\big(W_{\gamma)}\,g\big),\tag{109}$$

where the second equal sign comes from $W_{(\beta}W_{\gamma)} = 0$. After some algebra using properties of supersymmetric covariant derivatives and the spinorial superfield $W_\alpha$, we find that

$$\mathrm{D}^\alpha \mathcal{J}^{(1)}_{\alpha\beta\gamma} = 2i\,\partial_{\beta\gamma}\big(W^2\,g\big) + 4i\,W^2\,(\partial_{\beta\gamma}\,g) - (\mathrm{D}_{(\beta}W^2)(\mathrm{D}_{\gamma)}g).\tag{110}$$

It is evident that this equation is not consistent with the expected conservation equation. In particular, the last two terms measure the failure of $\mathcal{J}^{(1)}_{\alpha\beta\gamma}$ to satisfy (90). In order to eliminate these two terms, we consider additional contributions to the supercurrent as illustrated in (105). In particular, we examine the following term

$$\mathcal{J}^{(2)}_{\alpha\beta\gamma} = W^2\,\partial_{(\alpha\beta}\mathrm{D}_{\gamma)}\big(W^2 h\big) = W^2\big(\partial_{(\alpha\beta}\mathrm{D}_{\gamma)}W^2\big)h.\tag{111}$$

The structure of this term is motivated by the fact that the last two terms of (110) depend only on $W^2$ and its derivatives. One factor of $W^2$ is written explicitly and the second factor resides inside derivatives of $g$. Moreover, notice that $\mathcal{J}^{(2)}_{\alpha\beta\gamma}$ has two equivalent representations due to the nilpotency condition of $W_\alpha$ (i.e. $W_\alpha W_\beta W_\gamma \equiv 0$). The factor $h$ will be determined by demanding the cancellation of the last two terms in (110).

   Direct calculation using (A.25) and (A.26) gives

$$\mathrm{D}^\alpha \mathcal{J}^{(2)}_{\alpha\beta\gamma} = 2i\,(\mathrm{D}_{(\beta}W^2)\mathrm{D}_{\gamma)}\mathrm{D}^2\big(W^2 h\big) + 8\,W^2\,\partial_{\beta\gamma}\,\mathrm{D}^2\big(W^2 h\big) + 6\,(\mathrm{D}^\alpha W^2)\,\partial_{\beta\gamma}\,\mathrm{D}_\alpha\big(W^2 h\big).\tag{112}$$

The last two terms are not independent; in fact they are related by the following identity

$$W^2\,\partial_{\beta\gamma}\mathrm{D}^2\big(W^2 h\big) + (\mathrm{D}^\alpha W^2)\partial_{\beta\gamma}\mathrm{D}_\alpha\big(W^2 h\big) = -(\mathrm{D}^2 W^2)\partial_{\beta\gamma}\big(W^2 h\big),\tag{113}$$

which follows from the nilpotency of $W_\alpha$ (see proof in A.2). Observe that if we choose

$$h = f'(T)\quad\Rightarrow\quad \frac{2}{\kappa^2}\,\mathrm{D}^2(W^2 h) = g - f(T),\tag{114}$$

then the first two terms in (112) have a similar structure as the anomalous terms in (110). With these substitutions, we find

$$\begin{aligned}\frac{2}{\kappa^2}\,\mathrm{D}^\alpha \mathcal{J}^{(2)}_{\alpha\beta\gamma} = &-6\,\partial_{\beta\gamma}\Big[T\,W^2\,f'(T)\Big] + 4\,W^2\,\partial_{\beta\gamma}f(T) + 2\,W^2\,\partial_{\beta\gamma}\,g\\ &+ 2i\,(\mathrm{D}_{(\beta}W^2)\big(\mathrm{D}_{\gamma)}g\big) - 2i\,(\mathrm{D}_{(\beta}W^2)\big(\mathrm{D}_{\gamma)}f(T)\big).\end{aligned}\tag{115}$$

Surprisingly, most of these terms are associated with each other via the following identity

$$(\mathrm{D}_{(\beta}W^2)(\mathrm{D}_{\gamma)}g) = -2i\,W^2\,\partial_{\beta\gamma}g + 4i\,W^2\,\partial_{\beta\gamma}f(T) - 2\big(\mathrm{D}_{(\beta}f(T)\big)\big(\mathrm{D}_{\gamma)}W^2\big).\tag{116}$$

We leave this derivation in Appendix A.2. Using (116), (115) can be rewritten as follows:

$$\frac{2}{\kappa^2}\,\mathrm{D}^\alpha \mathcal{J}^{(2)}_{\alpha\beta\gamma} = -6\,\partial_{\beta\gamma}\Big[T\,W^2\,f'(T)\Big] + 4\,W^2\,\partial_{\beta\gamma}\,g + i\,(\mathrm{D}_{(\beta}W^2)\big(\mathrm{D}_{\gamma)}g\big).\tag{117}$$

Finally, we combine (110) and (117) and we find the following conservation equation

$$\mathrm{D}^\alpha\Big[\,\mathcal{J}^{(1)}_{\alpha\beta\gamma} - i\,\frac{2}{\kappa^2}\,\mathcal{J}^{(2)}_{\alpha\beta\gamma}\Big] = 2i\,\partial_{\beta\gamma}\Big[W^2 f(T) + 4\,W^2\,T\,f'(T)\Big].\tag{118}$$

From this we can immediately identify the supercurrent multiplet of the Maxwell-Goldstone multiplet,

$$\mathcal{J}_{\alpha\beta\gamma} = W_{(\alpha}\mathrm{D}_\beta\big(W_{\gamma)}\,g\big) - i\,\frac{2}{\kappa^2}\,W^2\,\partial_{(\alpha\beta}\mathrm{D}_{\gamma)}\big(W^2 f'(T)\big),\tag{119a}$$

$$\mathcal{J} = W^2 f(T) + 4\,W^2\,T\,f'(T).\tag{119b}$$

## 3.3 Flow operator as supercurrent-squared operator

The supercurrent multiplet (119a, 119b) calculated above allows us to answer the question we asked at the beginning of section 3. In particular, if we view the MG theory as the result of a flow triggered by the operator $\mathcal{O}_{\kappa^2}$ of (87), then can we give a $T^2$-like interpretation to this flow? In other words, can $\mathcal{O}_{\kappa^2}$ be expressed as a linear combination of "*supercurrent-squared*" terms?

**Supercurrent-squared scalars**    To answer this question, we search for superspace operators defined in terms of the supercurrent and supertrace and their derivatives such that their bosonic truncations will produce $T^2$ terms. In general, there are three types of $T^2$ terms: (1.) the square of the traceless part of the stress tensor $\widehat{T}^{\underline{ab}}\widehat{T}_{\underline{ab}}$; (2.) the square of the trace $\Theta^2$; (3.) the mixed term formed from the product of the trace with the "strength" of the traceless part $\Theta\sqrt{\widehat{T}^{\underline{ab}}\widehat{T}_{\underline{ab}}}$. Recall that the traceless part is a component of the supercurrent, whereas the trace resides in supertrace (96). Therefore, a natural proposal for the superspace operators in question is the following:

$$
\begin{aligned}
\mathcal{O}_{T^2} &= J^{\alpha\beta\gamma}J_{\alpha\beta\gamma}\,, \\
\mathcal{O}_{\Theta^2} &= J\,\mathrm{D}^2 J\,, \\
\mathcal{O}_{\Theta R} &= J\sqrt{\mathrm{D}^{(\alpha}J^{\beta\gamma\delta)}\mathrm{D}_{(\alpha}J_{\beta\gamma\delta)}}\,.
\end{aligned}
\tag{120}
$$

These operators are certainly not unique, however they are computationally convenient. They partition the three types of contributions mentioned above based on the fact that the traceless and trace parts of the energy-momentum tensor reside in different superfields. By integrating these operators over the fermionic directions and using (92), we find their corresponding spacetime components to be

$$
\begin{aligned}
\int d^2\theta\,\mathcal{O}_{T^2} &= -t^{\rho\alpha\beta\gamma}(x)t_{\rho\alpha\beta\gamma}(x) + \text{fermions}, \\
\int d^2\theta\,\mathcal{O}_{\Theta^2} &= \frac{1}{9}\Theta(x)^2 + \text{fermions}, \\
\int d^2\theta\,\mathcal{O}_{\Theta R} &= \frac{4!}{3}\Theta(x)\sqrt{t^{\alpha\beta\gamma\delta}(x)t_{\alpha\beta\gamma\delta}(x)} + \text{fermions}.
\end{aligned}
\tag{121}
$$

**Flow operator**    Now we evaluate these operators for MG theory. In (120) we plug in (119a) and (119b) to find

$$
\mathcal{O}_{T^2} = -4!2\,\kappa^2\,W^2\,T\left[f(T)+Tf'(T)\right]^2\,,
\tag{122a}
$$

$$
\mathcal{O}_{\Theta^2} = \frac{\kappa^2}{2}W^2\,T\left[f(T)+4\,T\,f'(T)\right]^2\,,
\tag{122b}
$$

$$
\mathcal{O}_{\Theta R} = 4!2\sqrt{6}\,\kappa^2\,T\,W^2\left[f(T)+4\,T\,f'(T)\right]\left[f(T)+Tf'(T)\right]\,,
\tag{122c}
$$

where we have used the nilpotency condition $W^3 \equiv 0$ repeatedly. It is straightforward to check that $f(T)$ defined as (85) satisfies the following constraint

$$
f'(T) = \frac{1}{2}f(T)^2 + T\,f(T)f'(T)\,.
\tag{123}
$$

With that in mind, we can show that the flow operator can be written as

$$
\begin{aligned}
\mathcal{O}_{\kappa^2} &= -\frac{1}{\kappa^2}W^2\,T\,f'(T) \\
&= \frac{1}{\kappa^4}\left[\frac{1}{108}\,\mathcal{O}_{T^2} + \frac{1}{9}\,\mathcal{O}_{\Theta^2} - \frac{1}{432\sqrt{6}}\,\mathcal{O}_{\Theta R}\right]\,.
\end{aligned}
\tag{124}
$$

This concludes the proof that the supersymmetric Maxwell-Goldstone theory satisfies a supercurrent-square like flow given by

$$\kappa^4 \frac{\partial \mathcal{L}_{\kappa^2}}{\partial \kappa^2} = \frac{1}{108} \, \mathcal{O}_{T^2} + \frac{1}{9} \, \mathcal{O}_{\Theta^2} - \frac{1}{432\sqrt{6}} \, \mathcal{O}_{\Theta R} \,. \tag{125}$$

It is important to emphasize that this superspace flow equation holds off-shell. In other words, equations (122) and (124) do not require the use of equations of motion.

**Bosonic truncation**  $\mathcal{O}_{\kappa^2}$ is a superspace operator that drives the flow in the space of manifestly $\mathcal{N} = 1$ supersymmetric theories. We can use it to define the operator that controls the flow of the bosonic truncation of the theory[12]

$$O_{\kappa^2}^{MG} := \kappa^4 \frac{9}{2} \left[ \int d^2\theta \, \mathcal{O}_{\kappa^2} \right]_{\text{fermions}\to 0} \,. \tag{126}$$

Using (121) and converting the pairs of spinorial indices to spacetime indices (98) we find

$$O_{\kappa^2}^{MG} = -\frac{1}{6} T^{\underline{ab}} T_{\underline{ab}} + \frac{1}{9} \Theta^2 - \frac{1}{9} \Theta \sqrt{\frac{3}{8} \widehat{T}^{\underline{ab}} \widehat{T}_{\underline{ab}}} \,, \tag{127}$$

which matches the operator $O_{\kappa^2}$ defined in (20).

### 3.4 Flow interpretations for tensor-Goldstone

The $3d$, $\mathcal{N} = 1$ vector multiplet has a *variant* description in terms of a dual superfield $U_\alpha$. This is demonstrated in [30] where this dual supermultiplet was used to provide an equivalent Goldstone multiplet corresponding to a nonlinear realization of the second broken supersymmetry. Therefore, the question of interpreting the nonlinear theory as the flow of a $T^2$-like operator could also be asked for this dual tensor-Goldstone description. In this section, we repeat the previous analysis in order to: (1.) find the appropriate supercurrent multiplet $\{\mathcal{J}_{\alpha\beta\gamma}, \mathcal{J}\}$ corresponding to the TG; (2.) use it to define the deformations of the theory that include $T^2$-like terms; (3.) show that the TG theory defines a flow operator $\mathcal{O}_{\tilde{\kappa}^2}$ which can be expressed as a linear combination of the above deformations; (4.) extract the bosonic truncation of this flow operator.

**Preliminaries**  We start with a brief reminder of the $3d$ variant description. The superspace action takes the following form.

$$S_{\text{TM}} \sim - \int d^3x \, d^2\theta \, U^2 \,, \tag{128}$$

where the superfield $U_\alpha$ is constrained by the following Bianchi Identity:

$$D^\alpha D^\beta U_\alpha = 0 \quad \Rightarrow \quad D^2 U_\alpha = -i \, \partial_\alpha{}^\beta U_\beta \,. \tag{129}$$

This can be solved by expressing $U_\alpha$ in terms of an unconstrained scalar superfield $G$.

$$U_\alpha = D_\alpha G \,. \tag{130}$$

In [30] we promoted this tensor multiplet to a tensor-Goldstone multiplet, consistent with the spontaneous breaking of $\mathcal{N} = 2 \to \mathcal{N} = 1$, described by the following action:

$$S_{\text{TG}} = - \int d^3x \, d^2\theta \, U^2 h(\tilde{T}) \,, \tag{131}$$

---

[12]The prefactor $\kappa^4$ provides the correct engineering dimensions for a $T^2$ deformation, while the numerical factor is a normalization choice.

where

$$h(x) = 1 - \frac{x}{1 + x + \sqrt{1+2x}} = f(-x), \quad \tilde{T} := \frac{2}{\tilde{\kappa}^2} \, \mathrm{D}^2 U^2 \,. \tag{132}$$

The dimensionful parameter $\tilde{\kappa}$ has the same engineering dimension as $\kappa$ ($[\tilde{\kappa}] = 3/2$) and describes the scale at which we break the second supersymmetry. Similar to the MG, the $\tilde{\kappa}$ dependence of the TG Lagrangian defines a curve which can be intepreted as a flow. This flow is triggered by the following operator

$$\mathcal{O}_{\tilde{\kappa}^2} = \frac{\partial \mathcal{L}_{TG}}{\partial \tilde{\kappa}^2} = \frac{1}{\tilde{\kappa}^2} \, U^2 \, \tilde{T} \, h'(\tilde{T}) \,. \tag{133}$$

Varying the action (131) with respect to the superfield $G$ yields the following equation of motion and its variants:

$$\begin{aligned}
\mathrm{D}^\rho \big\{ U_\rho \, \tilde{g} \big\} = 0 \quad &\Longleftrightarrow \quad \mathrm{D}_\alpha \big\{ U_\beta \, \tilde{g} \big\} = \mathrm{D}_\beta \big\{ U_\alpha \, \tilde{g} \big\} \\
&\Longrightarrow \quad \mathrm{D}^2 \big[ U^\alpha \, \tilde{g} \big] = i \, \partial^{\alpha\beta} \big[ U_\beta \, \tilde{g} \big] \,,
\end{aligned} \tag{134}$$

where the factor $\tilde{g}$ is defined as

$$\tilde{g} := h(\tilde{T}) + \frac{2}{\tilde{\kappa}^2} \mathrm{D}^2 \big[ U^2 h'(\tilde{T}) \big] \,. \tag{135}$$

Moreover, using properties of superspace covariant derivatives and $U_\alpha$ nilpotency condition ($U^3 \equiv 0$), we find the following identity:

$$(\mathrm{D}_{(\beta} U^2)(\mathrm{D}_{\gamma)} \tilde{g}) = -2i \, U^2 \, \partial_{\beta\gamma} \tilde{g} + 4i \, U^2 \, \partial_{\beta\gamma} h(\tilde{T}) - 2 \big( \mathrm{D}_{(\beta} h(\tilde{T}) \big) \big( \mathrm{D}_{\gamma)} U^2 \big) \,. \tag{136}$$

This is an analog of (116) for the TG.

**Comparison between MG and TG** Notice that the Lagrangians of the MG and TG have similar structures ($W^\alpha \mapsto U^\alpha$, $T \mapsto \tilde{T}$, $f(T) \mapsto h(\tilde{T})$). However, we would like to emphasize the following differences (see table 1).

In the free theory limit $\kappa, \tilde{\kappa} \to \infty$ ($g, \tilde{g} \to 1$), the Bianchi identity of one becomes the EoM of the other. This is another way to demonstrate the duality between these two theories. But for the interacting theories ($g, \tilde{g} \neq 1$), there is a mismatch. The conservation of the MG supercurrent multiplet relies on the Bianchi identity of $W_\alpha$ and EoM (left column). However, these identities are modified in the TG case (right column). In order to compute the supercurrent-squared operators $\{\mathcal{O}_{T^2}, \mathcal{O}_{\Theta^2}, \mathcal{O}_{\Theta R}\}$ for the TG multiplet using similar arguments as those in section 3.3, we use the following property

$$U^2 \, \tilde{g} \big( \mathrm{D}_\alpha U_\beta \big) = U^2 \, \mathrm{D}_\alpha \big( U_\beta \, \tilde{g} \big) = U^2 \, \mathrm{D}_\beta \big( U_\alpha \tilde{g} \big) = U^2 \, \tilde{g} \big( \mathrm{D}_\beta U_\alpha \big) \,. \tag{137}$$

Namely, in the presence of an additional $U^2$ factor, $\tilde{g}$ can move in and out of the spinorial derivative in the EoM (134), due to the nilpotency property of $U_\alpha$.

Table 1: Comparisons of MG and TG Bianchi identities and EoMs.

| MG | TG |
|---|---|
| Bianchi: $\mathrm{D}_\alpha W_\beta = \mathrm{D}_\beta W_\alpha$ | EoM: $\mathrm{D}_\alpha \big( U_\beta \, \tilde{g} \big) = \mathrm{D}_\beta \big( U_\alpha \tilde{g} \big)$ |
| EoM: $\mathrm{D}^\alpha \mathrm{D}^\beta \big( W_\alpha \, g \big) = 0$ | Bianchi: $\mathrm{D}^\alpha \mathrm{D}^\beta U_\alpha = 0$ |

**Supercurrents**    Motivated by (109) and (111), we consider the following ansatz[13]

$$\mathscr{J}^{(1)}_{\alpha\beta\gamma} = U_{(\alpha} \tilde{g} \left( D_\beta U_{\gamma)} \right) \overset{U_{(\beta} U_{\gamma)}=0}{=} U_{(\alpha} D_\beta \left( U_{\gamma)} \tilde{g} \right), \tag{138}$$

$$\mathscr{J}^{(2)}_{\alpha\beta\gamma} = U^2 \partial_{(\alpha\beta} D_{\gamma)} \left( U^2 h'(\tilde{T}) \right) \overset{U^3 \equiv 0}{=} U^2 \left( \partial_{(\alpha\beta} D_{\gamma)} U^2 \right) h'(\tilde{T}). \tag{139}$$

Using the various properties of $U_\alpha$ (129, 134, 136, A.31, A.32) and the supersymmetric covariant derivative algebra (A.25, A.26), we find the following superspace conservation equation

$$D^\alpha \left[ \mathscr{J}^{(1)}_{\alpha\beta\gamma} + i \frac{2}{\tilde{\kappa}^2} \mathscr{J}^{(2)}_{\alpha\beta\gamma} \right] = 2i \partial_{\beta\gamma} \left( U^2 h(\tilde{T}) - 2 \tilde{T} U^2 h'(\tilde{T}) \right). \tag{140}$$

From this, we identify the supercurrent multiplet for tensor-Goldstone theory as follows:

$$\mathscr{J}_{\alpha\beta\gamma} = U_{(\alpha} \tilde{g} \left( D_\beta U_{\gamma)} \right) + i \frac{2}{\tilde{\kappa}^2} U^2 \partial_{(\alpha\beta} D_{\gamma)} \left( U^2 h'(\tilde{T}) \right), \tag{141a}$$

$$\mathscr{J} = U^2 \left[ h(\tilde{T}) - 2 \tilde{T} h'(\tilde{T}) \right]. \tag{141b}$$

**Supercurrent-squared operator**    Using the above supercurrent and supertrace, we evaluate the supercurrent-squared operators (120)

$$\mathcal{O}_{T^2} \approx -4! 2 \tilde{\kappa}^2 U^2 \tilde{T} \left[ h(\tilde{T}) + \tilde{T} h'(\tilde{T}) \right]^2, \tag{142a}$$

$$\mathcal{O}_{\Theta^2} = \frac{\tilde{\kappa}^2}{2} \tilde{T} U^2 \left[ h(\tilde{T}) - 2 \tilde{T} h'(\tilde{T}) \right]^2, \tag{142b}$$

$$\mathcal{O}_{\Theta R} \approx 4! 2 \sqrt{6} \tilde{\kappa}^2 U^2 \tilde{T} \left[ h(\tilde{T}) - 2 \tilde{T} h'(\tilde{T}) \right] \left[ h(\tilde{T}) + \tilde{T} h'(\tilde{T}) \right]. \tag{142c}$$

The '$\approx$' symbol signifies that the equality holds only on-shell. This is because equation (137) was used in the derivation of (142). Again, this is a consequence of the duality between the MG and TG descriptions. What was a Bianchi identity for $W_\alpha$ now becomes an equation of motion for $U_\alpha$. Off-shell, the expressions for $\mathcal{O}_{T^2}$ and $\mathcal{O}_{\Theta R}$ acquire corrections proportional to $\mathcal{H} := D^\gamma U_\gamma$ (for details see section A.3). It is straightforward to see that the equation of motion of $U_\alpha$ [ $D^\alpha(U_\alpha g) = 0$ ] is equivalent to $U^2 \tilde{g} \mathcal{H} = 0 \Rightarrow \mathcal{H} = 0$. As a consequence all these corrections vanish on-shell and also the auxiliary component $H := \mathcal{H}|$ vanishes. This is in agreement with the use of the on-shell equation $H = 0$ in section 2.2. Similar conditions were used in the analysis of supersymmetric flows in other dimensions; see, for example, [29].

Note that $h(\tilde{T})$ defined in (132) satisfies the following relation

$$-h'(\tilde{T}) = \frac{1}{2} h^2(\tilde{T}) + \tilde{T} h(\tilde{T}) h'(\tilde{T}). \tag{143}$$

Therefore, the superspace flow operator $\mathcal{O}_{\tilde{\kappa}^2}$ defined in (133) can be written on-shell as

$$\mathcal{O}_{\tilde{\kappa}^2} \approx \frac{1}{\tilde{\kappa}^4} \left[ \frac{1}{108} \mathcal{O}_{T^2} + \frac{1}{9} \mathcal{O}_{\Theta^2} - \frac{1}{432\sqrt{6}} \mathcal{O}_{\Theta R} \right]. \tag{144}$$

This expression confirms that the TG action (131) which describes the partial $\mathcal{N} = 2 \to \mathcal{N} = 1$ supersymmetry breaking can be understood as the result of a supercurrent-squared flow. We also stress that the operator above coincides precisely with the second line of (124) up to exchanging $\tilde{\kappa}^2$ with $\kappa^2$. However, unlike the MG case, this flow is an on-shell one.

---

[13]One can also motivate this ansatz by starting with (88) and performing the duality procedure described in [30].

**Bosonic truncation** As mentioned previously, the tensor-Goldstone multiplet is a variant description, and hence its bosonic truncation will be similar in form to the truncation of the Maxwell-Goldstone multiplet. In other words, the bosonic sectors of both Lagrangians take the form of $3d$ Born-Infeld. As a result, we expect the operators that trigger the corresponding flows to take identical forms. Indeed, if we define the spacetime flow operator corresponding to the TG analogously to (126), we find

$$
\begin{aligned}
O_{\kappa^2}^{TG} &:= \tilde{\kappa}^4 \frac{9}{2} \left[ \int d^2\theta \, \mathscr{O}_{\tilde{\kappa}^2} \right]_{\text{fermions} \to 0} \\
&= -\frac{1}{6} T^{\underline{ab}} T_{\underline{ab}} + \frac{1}{9} \Theta^2 - \frac{1}{9} \Theta \sqrt{\frac{3}{8} \widehat{T}^{\underline{ab}} \widehat{T}_{\underline{ab}}} \, .
\end{aligned}
\tag{145}
$$

As a function of the energy-momentum tensor, $O_{\kappa^2}^{TG}$ is identical to $O_{\kappa^2}^{MG}$. However, we remind the reader that the evaluation of $T_{\underline{ab}}$ differs for the two theories.

# 4 Derivation of supercurrents from $\mathcal{N} = 1$ supergravity

In this section, we repeat the derivation of the supercurrents for the models analysed in the previous section by using the superspace supergravity results of [91] and [92]. We stress that the notations in this section are the ones of [91] and differ from the ones employed in the previous section. For the reader's convenience, the conventions used in this section together with a map between the notation in this and the previous section is given in Appendix B. The analysis in this section is a manifestly supersymmetric extension of the definition of the Hilbert stress-energy tensor for a matter system. The reader can find standard background material on the subject in [81, 93]. Before describing the calculation of the supercurrents for vector and scalar models, we introduce in the next subsection the necessary building blocks.

## 4.1 Building blocks from superspace supergravity

We are interested in coupling matter systems to $3d$ $\mathcal{N} = 1$ Poincaré supergravity. The latter can be described as $3d$ $\mathcal{N} = 1$ conformal supergravity coupled to a conformal compensator scalar multiplet. In superspace, the conformal supergravity multiplet can efficiently be described by a curved superspace geometry with the torsion and curvature satisfying an appropriate set of constraints. In this paper, we will employ the so-called $SO(1)$ geometry for conformal supergravity – see [81, 91] and also [92, 94, 95] for related approaches. The superspace geometry is described in terms of covariant derivatives taking the following form

$$
\mathcal{D}_A = (\mathcal{D}_a, \mathcal{D}_\alpha) = E_A - \Omega_A \,,
\tag{146}
$$

where $E_A = E_A{}^M \partial / \partial z^M$ defines the inverse vielbein, and[14]

$$
\Omega_A = \frac{1}{2} \Omega_A{}^{bc} M_{bc} = -\Omega_A{}^b M_b = \frac{1}{2} \Omega_A{}^{\beta\gamma} M_{\beta\gamma} \,,
\tag{147}
$$

is the Lorentz connection. The Lorentz generators act on the covariant derivatives as follows:

$$
[M_{\alpha\beta}, \mathcal{D}_\gamma] = \varepsilon_{\gamma(\alpha} \mathcal{D}_{\beta)} \,, \qquad [M_{ab}, \mathcal{D}_c] = 2\eta_{c[a} \mathcal{D}_{b]} \,.
\tag{148}
$$

---

[14]The Lorentz generators with two vector indices ($M_{ab} = -M_{ba}$), with one vector index ($M_a$) and with two spinor indices ($M_{\alpha\beta} = M_{\beta\alpha}$) are related to each other by the rules: $M_a = \frac{1}{2} \varepsilon_{abc} M^{bc}$ and $M_{\alpha\beta} = (\gamma^a)_{\alpha\beta} M_a$. We will also use $\mathcal{D}_{\alpha\beta} = (\gamma^a)_{\alpha\beta} \mathcal{D}_a$ for the vector covariant derivative.

The torsions and curvatures associated with the geometry arising from (anti-)commutation relations of the covariant derivatives satisfy an appropriate set of conventional constraints as described in [91]. The resulting algebra is:

$$\{\mathcal{D}_\alpha, \mathcal{D}_\beta\} = 2i\mathcal{D}_{\alpha\beta} - 4i\mathcal{S}M_{\alpha\beta}, \tag{149a}$$

$$[\mathcal{D}_{\alpha\beta}, \mathcal{D}_\gamma] = -2\varepsilon_{\gamma(\alpha}\mathcal{S}\mathcal{D}_{\beta)} + 2\varepsilon_{\gamma(\alpha}\mathcal{C}_{\beta)\delta\rho}M^{\delta\rho} + \frac{2}{3}(\mathcal{D}_\gamma\mathcal{S})M_{\alpha\beta} - \frac{8}{3}(\mathcal{D}_{(\alpha}\mathcal{S})M_{\beta)\gamma}, \tag{149b}$$

$$[\mathcal{D}_a, \mathcal{D}_b] = -\frac{i}{2}\varepsilon_{abc}(\gamma^c)^{\alpha\beta}\left\{\mathcal{C}_{\alpha\beta\gamma}\mathcal{D}^\gamma + \frac{4}{3}(\mathcal{D}_\alpha\mathcal{S})\mathcal{D}_\beta - (\mathcal{D}_{(\alpha}\mathcal{C}_{\beta\gamma\delta)})M^{\gamma\delta} + \frac{2}{3}(\mathcal{D}^2\mathcal{S} - 6i\mathcal{S}^2)M_{\alpha\beta}\right\}, \tag{149c}$$

where the torsion superfields $\mathcal{S}$ and $\mathcal{C}_{\alpha\beta\gamma}$ satisfy the Bianchi identity

$$\mathcal{D}^\gamma\mathcal{C}_{\alpha\beta\gamma} = -\frac{4i}{3}\mathcal{D}_{\alpha\beta}\mathcal{S}. \tag{150}$$

The gauge group of conformal supergravity consists of (i) superspace general coordinate and local Lorentz transformations; (ii) super-Weyl transformations. Covariant general coordinate transformations and local structure group transformations act on the covariant derivatives and on a generic tensor superfield $U$ belonging to some representation of the Lorentz groups as

$$\delta_\mathcal{K}\mathcal{D}_A = [\mathcal{K}, \mathcal{D}_A], \quad \mathcal{K} = \xi^B(z)\mathcal{D}_B + \frac{1}{2}K^{bc}(z)M_{bc}, \quad \delta_\mathcal{K}U = \mathcal{K}U, \tag{151}$$

with the gauge parameters $\xi^B$ and $K^{bc}$ obeying natural reality conditions but otherwise arbitrary. Super-Weyl transformations, which are related to local dilatation and special conformal transformations, arise as an invariance of the superspace geometry's conventional constraints. In particular, the algebra (149) is invariant under the following super-Weyl transformations of the covariant derivatives and torsion superfields

$$\delta_\sigma\mathcal{D}_\alpha = \frac{1}{2}\sigma\mathcal{D}_\alpha + (\mathcal{D}^\beta\sigma)M_{\alpha\beta}, \tag{152a}$$

$$\delta_\sigma\mathcal{D}_a = \sigma\mathcal{D}_a + \frac{i}{2}(\gamma_a)^{\alpha\beta}(\mathcal{D}_\alpha\sigma)\mathcal{D}_\beta + \varepsilon_{abc}(\mathcal{D}^b\sigma)M^c, \tag{152b}$$

$$\delta_\sigma\mathcal{S} = \sigma\mathcal{S} - \frac{i}{4}\mathcal{D}^2\sigma, \qquad \delta_\sigma\mathcal{C}_{\alpha\beta\gamma} = \frac{3}{2}\sigma\mathcal{C}_{\alpha\beta\gamma} - \frac{1}{2}\mathcal{D}_{(\alpha\beta}\mathcal{D}_{\gamma)}\sigma, \tag{152c}$$

with the parameter $\sigma$ being a real but otherwise unconstrained scalar superfield.

Among tensor superfields $U$, a special role is played by *primary* ones. A primary superfield of weight $w$ transforms homogenously under super-Weyl transformations:

$$\delta_\sigma U = w\sigma U. \tag{153}$$

As mentioned before, Poincaré supergravity can be defined as conformal supergravity coupled to a Lorentz scalar compensating multiplet. For $3d$ $\mathcal{N} = 1$ this can be chosen to be a nowhere vanishing real primary scalar superfield $\varphi$ such that

$$\delta_\sigma\varphi = \frac{1}{2}\sigma\varphi, \quad \varphi \neq 0. \tag{154}$$

Another key ingredient in the formulation of general $3d$ $\mathcal{N} = 1$ supergravity-matter systems is the full superspace action principle. This is given by[15]

$$S = i\int d^{3|2}z\, E\,\mathcal{L}, \qquad E^{-1} = \text{Ber}(E_A{}^M). \tag{155}$$

---

[15]In the notation of [91] that we employ in this section, the superspace measure $d^{3|2}z := d^3x\,d^2\theta$ is purely imaginary. For this reason, we use a factor of $i$ in (155).

The superspace Lagrangian $\mathcal{L}$ is chosen to be a primary, weight 2 real scalar superfield, $\delta_\sigma \mathcal{L} = 2\sigma \mathcal{L}$. Thanks to the fact that $\delta_\sigma E = -2\sigma E$, the action in eq. (155) is invariant under super-Weyl transformations. As we will see in the examples of the coming subsections, in general, the Lagrangian $\mathcal{L}$ will be constructed out of matter fields, their covariant derivatives, and, for non-conformal models, the compensator $\varphi$.

As discussed in [81,92], the constraints associated with the conformal supergravity geometry of (149) and (150) can be solved in terms of a single conformal real prepotential superfield $\Psi_{\alpha\beta\gamma} = \Psi_{(\alpha\beta\gamma)}$. For the scope of our paper, it suffices to know the covariant derivatives and torsion superfields at first order in $\Psi$ about a flat Minkowski superspace geometry. Such results can be readily read off from the analysis given in Appendix B of [92]. One has

$$\mathcal{D}_\alpha = D_\alpha + i\Psi_{\alpha\gamma\delta}\partial^{\gamma\delta} - \frac{1}{4}(D^2\Psi_{\alpha\beta\gamma})M^{\beta\gamma} - \frac{i}{2}(\partial_{(\alpha}{}^\delta\Psi_{\beta\gamma)\delta})M^{\beta\gamma} - \frac{2i}{3}(\partial_{\beta\gamma}\Psi^{\beta\gamma\delta})M_{\delta\alpha} + \mathcal{O}(\Psi^2),$$
(156a)

$$
\begin{aligned}
\mathcal{D}_{\alpha\beta} = {}& \partial_{\alpha\beta} + (D_{(\alpha}\Psi_{\beta\gamma\delta)})\partial^{\gamma\delta} + \frac{1}{2}(D^\delta\Psi_{\delta\rho(\alpha})\partial_{\beta)}{}^\rho \\
& + \Big[ -\frac{i}{4}D^2\Psi_{\alpha\beta\gamma} + \frac{1}{2}\partial_{(\alpha}{}^\delta\Psi_{\beta\gamma)\delta} - \frac{1}{3}\partial^{\rho\delta}\Psi_{\rho\delta(\alpha}\varepsilon_{\beta)\gamma}\Big]D^\gamma \\
& + \Big[ -\frac{5}{4}\partial_{(\alpha}{}^\rho D_\beta\Psi_{\gamma\delta\rho)} + \frac{1}{4}\partial_{(\alpha\beta}D^\tau\Psi_{\gamma\delta)\tau} + \frac{1}{4}\partial^{\rho\tau}\Big(D_{(\tau}\Psi_{\gamma\rho\alpha)}\varepsilon_{\beta\delta} + D_{(\tau}\Psi_{\gamma\rho\beta)}\varepsilon_{\alpha\delta}\Big) \\
& \quad - \frac{1}{12}\varepsilon_{\gamma(\alpha}\varepsilon_{\beta)\delta}\partial^{\rho\epsilon}D^\tau\Psi_{\rho\tau\epsilon} + \frac{1}{8}\Big(\partial_{(\alpha}{}^\rho\varepsilon_{\beta)(\gamma}D^\tau\Psi_{\delta)\rho\tau} - \partial_{(\gamma}{}^\rho\varepsilon_{\delta)(\alpha}D^\tau\Psi_{\beta)\rho\tau}\Big)\Big]M^{\gamma\delta} \\
& + \mathcal{O}(\Psi^2),
\end{aligned}
$$
(156b)

$$\mathcal{S} = -\frac{1}{8}D^{(\alpha}\partial^{\beta\gamma)}\Psi_{\alpha\beta\gamma} + \mathcal{O}(\Psi^2),$$
(156c)

$$\mathcal{C}_{\alpha\beta\gamma} = \frac{1}{4}\partial_{(\alpha}{}^\delta D^2\Psi_{\beta\gamma)\delta} + \frac{i}{2}\partial_{(\alpha}{}^\delta\partial_\beta{}^\rho\Psi_{\gamma)\delta\rho} + \mathcal{O}(\Psi^2).$$
(156d)

Here $D_A = (\partial_a, D_\alpha)$ denote the flat Minkowski superspace covariant derivatives satisfying the algebra

$$\{D_\alpha, D_\beta\} = 2i\partial_{\alpha\beta}, \quad [\partial_a, D_\beta] = [\partial_a, \partial_b] = 0.$$
(157)

Note that the value of the compensator in the flat background is chosen to be a constant; in particular, we will choose $\varphi = 1$. Another useful result, which is implied by (156), is that the Berezinian around a flat background satisfies

$$E = 1 + \mathcal{O}(\Psi^2).$$
(158)

As discussed in [92], to obtain the previous representation, part of the supergravity gauge group, including super-Weyl transformations, has to be used and it is fixed. The residual gauge freedom of $\mathcal{D}_A$, $\mathcal{S}$, and $\mathcal{C}_{\alpha\beta\gamma}$ in (156) is described by transformations of the form

$$\delta\mathcal{D}_A := [\mathcal{K}, \mathcal{D}_A] + \delta_\sigma\mathcal{D}_A,$$
(159)

where the parameters $\xi^\alpha$, $K^{bc}$ and $\sigma$ of (151) and (152) should be functions of $\xi^a$ and its covariant derivatives:

$$\xi_\alpha = -\frac{i}{6}D^\beta\xi_{\beta\alpha} + \mathcal{O}(\Psi),$$
(160a)

$$K_{\alpha\beta} = 2D_{(\alpha}\xi_{\beta)} + \mathcal{O}(\Psi),$$
(160b)

$$\sigma = D_\alpha\xi^\alpha + \mathcal{O}(\Psi) = \frac{1}{3}\partial^a\xi_a + \mathcal{O}(\Psi).$$
(160c)

The prepotential superfield then transforms as

$$\delta\Psi_{\alpha\beta\gamma} = \frac{1}{2}D_{(\alpha}\xi_{\beta\gamma)} + \mathcal{O}(\Psi), \tag{161}$$

while the compensator $\varphi$ satisfies

$$\delta\varphi = -\frac{1}{12}\partial^{\alpha\beta}\xi_{\alpha\beta} + \mathcal{O}(\Psi). \tag{162}$$

Consider now a supergravity-matter system described by the full superspace action $S[U, \mathcal{D}_A, \varphi]$, and depending in general upon matter multiplets (that we denote by $U$), the conformal geometry, and the conformal compensator $\varphi$. Assuming the equations of motion for the matter superfields $U$ are satisfied, $\delta S / \delta U = 0$, the variation of the action induced by an infinitesimal deformation of the gravitational superfields $\Psi_{\alpha\beta\gamma}$ and the compensator $\varphi$ gives

$$\delta S[U, \mathcal{D}_A, \varphi] = i\int d^{3|2}z \left(\delta\Psi^{\alpha\beta\gamma}J_{\alpha\beta\gamma} + \delta\varphi J\right), \tag{163}$$

where we are taking a variation around a flat background, and hence we are dropping the $E$ factor in the final expression.

The superfields $J_{\alpha\beta\gamma}$ and $J$ define the supercurrent multiplet and, as we will see in examples in the coming subsections, can be computed explicitly by using the building blocks described above. The variation (163) must vanish if $\delta\Psi^{\alpha\beta\gamma}$ and $\delta\varphi$ are the gauge transformations (161) and (162). Since the gauge parameter $\xi_{\beta\gamma} = (\gamma^a)_{\beta\gamma}\xi_a$ in (161) and (162) is an arbitrary superfield, one concludes that the following supercurrent conservation equation must hold

$$D^\gamma J_{\alpha\beta\gamma} = \frac{i}{6}\partial_{\alpha\beta}J, \tag{164}$$

provided the equations of motion for the matter multiplets $U$ derived from the action $S[U, \mathcal{D}_A, \varphi]$ are satisfied. Note that if the model described by $S$ is superconformal, then the dependence upon the compensator $\varphi$ should disappear, $S = S[U, \mathcal{D}_A]$. In this case, the trace multiplet superfield vanishes, $J = 0$, and (164) simplifies to $D^\gamma J_{\alpha\beta\gamma} = 0$.

## 4.2 Calculation of supercurrents for vector models

In this subsection we are going to use the supergravity building block of the previous subsection to compute the supercurrent multiplet for vector multiplet models taking the following form

$$S = -\frac{1}{2}\int d^{3|2}z\, W^2 f(T), \quad W^2 = W^\alpha W_\alpha, \quad T := \frac{1}{8}D^2 W^2, \quad D^2 = D^\alpha D_\alpha, \tag{165}$$

for any function $f(T)$. Here $W_\alpha$ is the field strength of an Abelian vector multiplet. In our notations, an Abelian vector multiplet can be described by a closed super 2-form $F^{(2)} = \frac{1}{2}E^B\wedge E^A F_{AB}$, $dF^{(2)} = 0$, with components

$$F_{\alpha\beta} = 0, \tag{166a}$$

$$F_{a\beta} = -i(\gamma_a)_{\beta\gamma}W^\gamma = -F_{\beta a}, \tag{166b}$$

$$F_{ab} = \frac{1}{2}\varepsilon_{abc}(\gamma^c)^{\alpha\beta}D_\alpha W_\beta = -F_{ba}, \tag{166c}$$

where the field strength $W_\alpha$ is real and satisfies

$$D^\alpha W_\alpha = 0, \quad (W_\alpha)^* = W_\alpha. \tag{167}$$

We can write the $\theta$-components of $W_\alpha$ as

$$W_\alpha = \lambda_\alpha + \theta^\beta f_{\alpha\beta} + \frac{i}{2}\theta^2 \partial_{\alpha\beta}\lambda^\beta\,, \tag{168a}$$

$$\lambda_\alpha := W_\alpha|_{\theta=0}\,, \quad f_{\alpha\beta} := D_\beta W_\alpha|_{\theta=0} = \frac{1}{2}\varepsilon^{abc}(\gamma_a)_{\alpha\beta}f_{bc}\,, \quad \partial_{[a}f_{bc]} = 0\,, \tag{168b}$$

where note that $f_{ab} = F_{ab}|_{\theta=0}$ with $F_{ab}$ being the top component of the closed superform $F^{(2)}$ in eq. (166). The free supersymmetric Maxwell theory is described by the following action

$$S = -\frac{1}{2}\int d^{3|2}z\, W^\alpha W_\alpha = \int d^3x\left(-\frac{1}{4}f^{ab}f_{ab} - \frac{i}{2}\lambda^\alpha\partial_{\alpha\beta}\lambda^\beta\right)\,, \tag{169}$$

which corresponds to (165) with $f(T) = 1$. Note also that the definition of the superfield $T$ is such that its lowest component coincides with the free Maxwell Lagrangian:

$$T|_{\theta=0} = -\frac{1}{4}f^{ab}f_{ab} - \frac{i}{2}\lambda^\alpha\partial_{\alpha\beta}\lambda^\beta\,. \tag{170}$$

The first step to compute the supercurrent for the previous models is to couple them to Poincaré supergravity. The covariant vector multiplet is defined by a weight-3/2 primary real spinor superfield strength $\mathcal{W}_\alpha$ satisfying

$$\mathcal{D}^\alpha \mathcal{W}_\alpha = 0\,, \quad \delta_\sigma \mathcal{W}_\alpha = \frac{3}{2}\sigma \mathcal{W}_\alpha\,, \tag{171}$$

which represents the curved extension of $W_\alpha$. To lift the model (165) to supergravity we would also like to have a primary version of $T$ and $f(T)$. A straightforward calculation shows that the following superfield is primary and weight zero:

$$\mathcal{T} = \frac{1}{8}\Big(\varphi^{-3}(\mathcal{D}^2\varphi^{-5}\mathcal{W}^2) - 2i\varphi^{-8}\mathcal{S}\mathcal{W}^2\Big)\,, \quad \delta_\sigma\mathcal{T} = 0\,, \tag{172}$$

where $\mathcal{D}^2 := \mathcal{D}^\alpha \mathcal{D}_\alpha$ and $\mathcal{W}^2 := \mathcal{W}^\alpha \mathcal{W}_\alpha$. This expression for $\mathcal{T}$ can be obtained by using the fact that, given a weight-1/2 scalar primary superfield, as for example $\varphi$, the combination $(\mathcal{D}^2 - 2i\mathcal{S})\varphi$ transforms homogenously with weight 3/2 under super-Weyl transformations, see [92]. Then any function $f(\mathcal{T})$ is also a weight-zero primary superfield which reduces to $f(T)$ when taking the flat superspace limit, $\mathcal{D}_A = D_A$ and $\varphi = 1$. Since $\delta_\sigma \mathcal{W}^2 = 3\sigma \mathcal{W}^2$, it is clear that the following action is an appropriate curved extension of (165)

$$S = -\frac{1}{2}\int d^{3|2}z\, E\, \varphi^{-2}\mathcal{W}^2 f(\mathcal{T})\,. \tag{173}$$

By noticing that the nilpotency condition $\mathcal{W}^2\mathcal{W}_\alpha = 0$ holds, it follows that the previous action can be written in the following simplified form

$$S = -\frac{1}{2}\int d^{3|2}z\, E\, \varphi^{-2}\mathcal{W}^2 f(\mathbf{T})\,, \quad \mathbf{T} := \frac{1}{8}\varphi^{-8}\mathcal{D}^2\mathcal{W}^2\,. \tag{174}$$

This is our starting point to compute the supercurrent for (165).

It is simple to compute the supertrace, $J$. It suffices to take the variation of the compensator $\varphi$ about the flat Minkowski background ($\mathcal{D}_A = D_A$, $\varphi = 1$). We will denote this variation as $\delta_\varphi$ where the compensator should be thought as $\varphi = 1 + \delta\varphi$. By using the following result

$$\delta_\varphi f(\mathbf{T}) = -8\delta\varphi\, \mathbf{T}f'(T)\,, \tag{175}$$

one obtains

$$\delta_\varphi S = -\frac{1}{2}\int d^{3|2}z \, \delta\varphi \, W^2\Big\{-2f(T)-8Tf'(T)\Big\}. \tag{176}$$

By comparing with the compensator variation in (163), we obtain

$$J = -iW^2\Big(f(T)+4Tf'(T)\Big). \tag{177}$$

The variation with respect to $\Psi_{\alpha\beta\gamma}$ is slightly more involved. First of all, we need to obtain the expression of the covariant field strength $\mathcal{W}_\alpha$ in terms of $W_\alpha$ and the conformal prepotential. A natural candidate is given by the following ansatz

$$\mathcal{W}_\alpha = W_\alpha + a(D^\beta\Psi_{\alpha\beta\gamma})W^\gamma + b\Psi_{\alpha\beta\gamma}D^\beta W^\gamma + \mathcal{O}(\Psi^2). \tag{178}$$

By imposing $\mathcal{D}^\alpha\mathcal{W}_\alpha = 0$ up to linear order in $\Psi$ with $\mathcal{D}_\alpha$ given by (156a), and by using $D^\alpha W_\alpha = 0$, one can check that the constant parameters in (178) are uniquely fixed to be

$$a = -b = 1 \quad \Longrightarrow \quad \mathcal{W}_\alpha = W_\alpha + (D^\beta\Psi_{\alpha\beta\gamma})W^\gamma - \Psi_{\alpha\beta\gamma}D^\beta W^\gamma + \mathcal{O}(\Psi^2). \tag{179}$$

This, together with (156a), implies

$$\mathcal{W}^2 = W^2 - 2\Psi^{\alpha\beta\gamma}W_\alpha D_\beta W_\gamma + \mathcal{O}(\Psi^2), \tag{180a}$$

$$\mathbf{T} = \varphi^{-8}\left[T - \frac{1}{4}D^2\big(\Psi^{\alpha\beta\gamma}W_\alpha D_\beta W_\gamma\big) - \frac{i}{8}D_\alpha\big(\Psi^{\alpha\beta\gamma}\partial_{\beta\gamma}W^2\big) + \frac{i}{8}\partial_{\alpha\beta}\big(\Psi^{\alpha\beta\gamma}D_\gamma W^2\big) + \mathcal{O}(\Psi^2)\right]. \tag{180b}$$

At this point, by using the results above together with (158), it is straightforward to take the variation of (174) with respect to $\Psi^{\alpha\beta\gamma}$ around $\mathcal{D}_A = D_A$ and $\varphi = 1$, which we denote as $\delta_\Psi$. After some integration by parts, one obtains

$$\delta_\Psi S = \int d^{3|2}z \, \delta\Psi^{\alpha\beta\gamma}\left\{i\Big[f(T)+\frac{1}{8}D^2\big(W^2 f'(T)\big)\Big]W_{(\alpha}D_\beta W_{\gamma)} - \frac{i}{8}W^2 f'(T)\partial_{(\alpha\beta}D_{\gamma)}W^2\right\}. \tag{181}$$

From this one can readily read off the expression for $J_{\alpha\beta\gamma}$. Summarising, the supergravity analysis leads to the following supercurrent multiplet for the model (165)

$$J_{\alpha\beta\gamma} = -ig(T)W_{(\alpha}D_\beta W_{\gamma)} - \frac{1}{8}W^2 f'(T)\,\partial_{(\alpha\beta}D_{\gamma)}W^2, \tag{182a}$$

$$J = -iW^2\Big(f(T)+4Tf'(T)\Big), \tag{182b}$$

with the superfield $g(T)$ defined by

$$g(T) = f(T) + \frac{1}{8}D^2\big(W^2 f'(T)\big). \tag{183}$$

Up to mapping notations according to Appendix B, and inserting factors of $\kappa^2$ by field redefinitions, the supercurrent coincides with the result of the previous section, eqs. (119).

## 4.3 Calculation of supercurrents for tensor/scalar multiplet models

We conclude this section by looking at the calculation of the supercurrent for models based on tensor multiplets. In the notation of this section, we look at models described by the following action principle

$$S = -\frac{1}{2}\int d^{3|2}z \, U^2 h(\tilde T), \tag{184}$$

where the superfields $U_\alpha$ and $\tilde{T}$ satisfy

$$D^\alpha D_\beta U_\alpha = 0\,, \quad (U_\alpha)^* = U_\alpha\,, \quad U^2 := U^\alpha U_\alpha\,, \quad \tilde{T} = \frac{1}{8}D^2 U^2\,. \tag{185}$$

The solution of the constraint for the tensor multiplet is given in terms of a spinor derivative of a real scalar superfield $\rho$: $U_\alpha = iD_\alpha\rho$. We will use this decomposition as the starting point to couple the multiplet to supergravity. In fact, we promote the scalar superfield $\rho$ to be a primary of weight zero. This implies that the curved version of $U_\alpha$, which we denote as $\mathcal{U}_\alpha$, is a primary of weight-$1/2$. More specifically, we use

$$\rho\,, \quad \delta_\sigma \rho = 0\,, \quad \mathcal{U}_\alpha := i\mathcal{D}_\alpha\rho\,, \quad \delta\mathcal{U}_\alpha = \frac{1}{2}\sigma\mathcal{U}_\alpha\,. \tag{186}$$

A locally supersymmetric and super-Weyl invariant version of the model (184) is described by the action

$$S = -\frac{1}{2}\int d^{3|2}z\, E\, \varphi^2 \mathcal{U}^2 h(\tilde{\mathbf{T}})\,, \tag{187}$$

where $\mathcal{U}^2 := \mathcal{U}^\alpha \mathcal{U}_\alpha$ and

$$\tilde{\mathbf{T}} = \frac{1}{8}\varphi^{-4}\mathcal{D}^2 \mathcal{U}^2\,. \tag{188}$$

Note that $\tilde{\mathbf{T}}$ is not a primary superfield, as it transforms inhomogeneously under super-Weyl transformations. However, it is simple to show that, due to the nilpotency $\mathcal{U}^2\mathcal{U}_\alpha = 0$, it satisfies $\mathcal{U}^2 \delta_\sigma \tilde{\mathbf{T}} = 0$ which suffices to make (187) super-Weyl invariant.

By following the same analysis of the previous subsection, it is now simple to obtain the supercurrent for (184) by taking the variation of (187) with respect to the conformal prepotential $\Psi_{\alpha\beta\gamma}$ and the conformal compensator $\varphi$. First of all, the $\mathcal{U}_\alpha$ superfield is independent of the compensator $\varphi$ and it is simple to show that, around the flat Minkowski superspace background ($\mathcal{D}_A = D_A$, $\varphi = 1$) one has $\delta_\varphi \tilde{\mathbf{T}} = -4\delta\varphi\tilde{T}$. Next, by using (156a), the fact that $\rho$ is a scalar superfield, and $\partial^{\alpha\beta} = -iD^{(\alpha}D^{\beta)}$, one can immediately obtain the following results

$$\mathcal{U}_\alpha = i\big(D_\alpha + i\Psi_{\alpha\gamma\delta}\partial^{\gamma\delta}\big)\rho + \mathcal{O}(\Psi^2)\,, \tag{189a}$$

$$\delta_\Psi \mathcal{U}_\alpha = \delta\Psi_{\alpha\beta\gamma}D^\beta U^\gamma\,, \tag{189b}$$

$$\delta_\Psi \mathcal{U}^2 = 2\delta\Psi^{\alpha\beta\gamma}U_\alpha D_\beta U_\gamma\,, \tag{189c}$$

$$\delta_\Psi \tilde{\mathbf{T}} = \frac{i}{8}\delta\Psi^{\alpha\beta\gamma}\partial_{\alpha\beta}D_\gamma U^2 + \frac{i}{8}\big(\partial_{\alpha\beta}\delta\Psi^{\alpha\beta\gamma}\big)D_\gamma U^2 - \frac{i}{8}D_\alpha\big(\delta\Psi^{\alpha\beta\gamma}\partial_{\beta\gamma}U^2\big)$$
$$+ \frac{1}{4}D^2\big(\delta\Psi^{\alpha\beta\gamma}U_\alpha D_\beta U_\gamma\big)\,. \tag{189d}$$

By using the equations above, together with $\delta_\Psi E = 0$, it is straightforward to take an arbitrary variation of (187) with respect of $\Psi^{\alpha\beta\gamma}$ and $\varphi$ which, after performing some superspace integration by parts to bring the result in the form (163), leads to the following supercurrent

$$J_{\alpha\beta\gamma} = iU_{(\alpha}\big(D_\beta U_{\gamma)}\big)\Big[h(\tilde{T}) + \frac{1}{8}D^2\big(U^2 h'(\tilde{T})\big)\Big] - \frac{1}{8}U^2\big(\partial_{(\alpha\beta}D_{\gamma)}U^2\big)h'(\tilde{T})\,, \tag{190a}$$

$$J = iU^2\Big[h(\tilde{T}) - 2\tilde{T}h'(\tilde{T})\Big]\,. \tag{190b}$$

Up to mapping notations according to Appendix B, and inserting factors of $\tilde{\kappa}^2$ by field redefinitions, the supercurrent coincides with the result of the previous section, eq. (141).

# 5 Conclusion

In this work, we have realized the effective description of a D2-brane via an irrelevant deformation of a free theory. The flow connecting this nonlinear description to the free point can be realized either in gauge theory variables or in the tensor-Goldstone presentation, where the bosonic field content is a scalar field that can be interpreted as the Hodge dual of the Born-Infeld gauge field.

When restricting to the bosonic truncation of these theories, the operator driving our flow is a dimension-6 combination of stress tensors which is similar to the $T\overline{T}$-like deformations which are known to produce string and brane actions in $d = 2$ and $d = 4$, respectively. However, we have found that in $d = 3$ we must introduce a new ingredient which is a non-analytic combination of energy-momentum tensors, $R = \sqrt{\frac{3}{8}\left(T^{ab}T_{ab} - \frac{1}{3}\left(T^a_a\right)^2\right)}$.

We have shown that this flow can be made manifestly supersymmetric by constructing the deforming operator directly from supercurrents in $\mathcal{N} = 1$ superspace, including a superfield version of the non-analytic operator $R$. For the deformation associated to the Maxwell-Goldstone multiplet, the flow equation holds fully off-shell, which is only known to occur in a handful of other cases [27, 52, 53]. More generally, both the Maxwell-Goldstone and tensor-Goldstone flows provide new examples, besides the known results in $d = 2$ and $d = 4$, where a deformation by a quadratic combination of supercurrents produces a theory with an additional non-linearly realized supersymmetry. We have also developed technology for computing the supercurrents in a fairly general class of vector and tensor models using two approaches: by solving the consistency conditions provided by the superspace conservation equations, and by coupling to $\mathcal{N} = 1$ supergravity then taking the linearized limit.

These observations provide further evidence for a deeper connection between deformations by conserved currents, theories of strings and branes, and spontaneously broken symmetries (including supersymmetries). The full scope of this connection remains to be explored, and there remain several interesting avenues for future research. Perhaps the most natural extension is to consider versions of the analysis in this work for theories with more supersymmetry and different symmetry breaking patterns. Below we outline some other directions, and we hope to return to some of these questions in future work.

*ModMax-like extensions and DBI*

Although this manuscript has focused on irrelevant deformations of free seed theories, there are also examples in $d = 2$ and $d = 4$ of stress tensor flows driven by marginal combinations. In the four-dimensional setting, a flow driven by the root-$T\overline{T}$-like combination

$$\frac{\partial \mathcal{L}}{\partial \gamma} = \frac{1}{2}\sqrt{T^{ab}T_{ab} - \frac{1}{4}\left(T^a_a\right)^2}, \tag{191}$$

produces the ModMax theory [44–46] when the initial condition is the ordinary Maxwell theory [47]. As we have mentioned in Section 1, this flow commutes with the irrelevant $T\overline{T}$-like flow which deforms the Maxwell Lagrangian into the Born-Infeld theory, and this pair of commuting flows can be used to construct a two-parameter family of ModMax-Born-Infeld theories (and their supersymmetrix extensions) [48, 49]. A similar pair of flows defines a collection of Modified-Nambu-Goto theories in two dimensions [17, 34, 50].

It would be interesting to investigate whether some analogue of these marginal flows exist for three-dimensional theories. It is straightforward to see that no such flow exists in $d = 3$ if the seed theory describes a single free gauge field or a single free scalar, since in both of these cases the deformed Lagrangian can depend on only a single Lorentz invariant (proportional to $f^{ab}f_{ab}$ or $\partial^a\phi\partial_a\phi$, respectively). As a consequence, the $3d$ analogue of the marginal flow

[(191)](191) simply re-scales the kinetic term for either of these theories, exactly as in the $2d$ case of [34].

One might evade this issue by beginning with a seed theory $\mathcal{L}_0 = -\frac{1}{4}f^{ab}f_{ab} - \partial^a\phi^i\partial_a\phi^i$ which contains both a gauge field and a collection of scalars $\phi^i$. This is natural from the perspective of a physical D2-brane, which supports both a gauge field on its worldvolume and several scalar fields which describe the transverse fluctuations of the brane, and whose dynamics are jointly described by the Dirac-Born-Infeld (DBI) theory. It is interesting to ask whether the full DBI Lagrangian can be obtained from a flow beginning with this seed theory, which would generalize the flows for a single scalar or gauge field which we considered in this work. Further, one could attempt to construct a two-parameter family of commuting flows to produce a Modified-Dirac-Born-Infeld theory which depends on both an irrelevant parameter $\lambda$ and a marginal parameter $\gamma$. It would be especially exciting if one could argue that this theory emerges as an effective description of a brane along with some other string theory ingredients which have the effect of turning on the marginal coupling $\gamma$. This could potentially provide deeper insight into the nature of ModMax-like interactions by allowing us to directly engineer such couplings in string theory.

*Connections to soft behavior*

We have seen around equation [(72)](72) that the Dirac action – by which we mean the scalar sector of the Dirac-Born-Infeld theory – can be obtained as an irrelevant deformation of a free scalar theory in any spacetime dimension $d$. On the other hand, there is a different characterization which completely determines this theory in the context of scattering amplitudes. It was shown in [96] that requiring a general theory $\mathcal{L}(\partial^a\phi\partial_a\phi)$ for a scalar field to exhibit enhanced soft behavior, in the sense that tree amplitudes vanish as $\mathcal{A}(p) = \mathcal{O}(p^2)$ as the momentum $p$ of any external leg is taken to zero, uniquely fixes the Lagrangian to take the square-root form of the Dirac theory.

The property that a scalar theory exhibit this enhanced soft behavior is therefore equivalent to the statement that the theory is obtained as an irrelevant deformation of a free scalar by the operator appearing in [(72)](72). It would be intriguing to explore whether there is a physical reason why *this* operator, in particular, is connected to soft limits. Because the soft behavior of the Dirac Lagrangian is a consequence of the non-linearly realized symmetry of this theory, which arises because embedding a brane in spacetime spontaneously breaks some of the ambient Poincaré symmetry, this question is another way of asking why exactly a deformation by an irrelevant combination of stress tensors causes a theory to develop an additional non-linearly realized symmetry.

It is also interesting to ask what principle replaces this enhanced soft behavior in the gauge theory case, where amplitudes are actually singular in the soft limit due to the Weinberg soft photon theorem. One possibility is to use multi-chiral soft limits or a combination of single soft limits and dimensional reduction, as in [97], which appears to also single out the Born-Infeld Lagrangian and thus may have a relationship with the $T\overline{T}$-like deformations which produce this theory in $d = 3$ and $d = 4$. Another possibility is that a generalization of the requirement that a stress tensor deformation preserve the absence of birefringence, which uniquely picks out the appropriate $T\overline{T}$-like deformation of gauge theories in four dimensions [48], will identify the appropriate operator in other contexts. There is no notion of birefringence in three spacetime dimensions because photons have only a single physical polarization, but one might hope that preserving a version of the zero-birefringence condition in higher dimensions or for $p$-form electrodynamics with $p > 2$ may pinpoint a distinguished stress tensor deformation in those contexts.

*Potential applications to holography*

Part of the motivation for the present work is the observation that a $T\overline{T}$ deformation of a $2d$ free scalar yields the Nambu-Goto Lagrangian [3], which hints at connections between stress tensor deformations and theories of strings and branes. Another such connection comes from the "single trace" version of the $T\overline{T}$ deformation whose holographic interpretation was proposed in [98–100] and whose further elucidated in [101–105].

This correspondence involves a solution of type IIB supergravity with a collection of fundamental strings and NS5-branes. In the near horizon region of both the strings and the five-branes, this spacetime approaches $AdS_3 \times S^3 \times T^4$, which admits a holographic description in terms of a two-dimensional conformal field theory. As one moves away from the deep bulk region, the gravity solution interpolates from an asymptotically $AdS_3$ spacetime to a linear dilaton spacetime. From the $CFT_2$ perspective, the procedure which recouples this linear dilaton region can be interpreted as an irrelevant deformation that is similar to $T\overline{T}$. In this context, a stress tensor deformation of a $CFT_2$ can be viewed holographically as "undoing" the process of taking a near-horizon limit, i.e. as "zooming out" from the deep bulk of a spacetime involving F1-strings.

It would be very interesting if this interpretation could be generalized to other gravity solutions. For instance, it might be that a "single-trace" version of the $3d$ stress tensor deformation considered in the present work might recouple the asymptotic region of some supergravity solution involving a stack of D2-branes, just as the single-trace $2d$ $T\overline{T}$ operator recouples an asymptotic region of a gravity solution involving F1-strings.

# Acknowledgments

C. F. is supported by U.S. Department of Energy grant DE-SC0009999 and by funds from the University of California. C.F. also thanks the University of Queensland for hospitality during a visit which led to some of the results in this work. Z.H. and G.T.M. thank Johannes Otto Faller for helpful discussions and the development of some notes on $3d$ supersymmetry.

**Funding information**    The research of Y. H. is supported by the Celestial Holography Initiative at the Perimeter Institute for Theoretical Physics. Research at the Perimeter Institute is supported by the Government of Canada through the Department of Innovation, Science and Industry Canada and by the Province of Ontario through the Ministry of Colleges and Universities. The work of K.K. is supported in part by the endowment of the Clark Leadership Chair in Science at the University of Maryland, College Park. K.K gratefully acknowledge the hospitality of the Physics Department at the University of Maryland, College Park. Z.H. is supported by a postgraduate scholarship at the University of Queensland. The work of G.T.-M. is supported by the Australian Research Council (ARC) Future Fellowship FT180100353, and by the Capacity Building Package of the University of Queensland.

# A    Conventions and identities for section 3

## A.1    Conventions

**Indices**    In three dimensions, we use underlined Latin letters to denote spacetime indices: $\underline{a} = 0, 1, 2$; and we use Greek letters for spinorial indices: $\alpha = 1, 2$.

**Gamma matrix**   We choose $3d$ Gamma matrices as

$$\begin{aligned}
(\gamma^0)_\alpha{}^\beta &= (i\sigma^2)_\alpha{}^\beta\,,\\
(\gamma^1)_\alpha{}^\beta &= (\sigma^3)_\alpha{}^\beta\,,\\
(\gamma^2)_\alpha{}^\beta &= (\sigma^1)_\alpha{}^\beta\,,
\end{aligned} \tag{A.1}$$

which satisfy the Clifford algebra:

$$\{\gamma^{\underline{a}},\gamma^{\underline{b}}\} = 2\eta^{\underline{ab}}\mathbb{I}\,, \tag{A.2}$$

where the Minkowski metric is

$$\eta_{\underline{ab}} = \eta^{\underline{ab}} = \begin{pmatrix} -1 & 0 & 0\\ 0 & 1 & 0\\ 0 & 0 & 1 \end{pmatrix}. \tag{A.3}$$

The gamma matrix has the following trace identity,

$$(\gamma^{\underline{a}})_\alpha{}^\beta (\gamma_{\underline{b}})_\beta{}^\alpha = 2\delta^{\underline{a}}_{\underline{b}}\,. \tag{A.4}$$

We use the spinor metric to raise and lower spinor indices:

$$\psi_\alpha = \psi^\beta C_{\beta\alpha}\,, \qquad \psi^\alpha = C^{\alpha\beta}\psi_\beta\,, \tag{A.5}$$

where the definition of the spinor metric is

$$C_{\alpha\beta} = -C_{\beta\alpha} = -C^{\alpha\beta} = \begin{pmatrix} 0 & -i\\ i & 0 \end{pmatrix}, \tag{A.6}$$

with the following identities

$$C_{\alpha\beta}C^{\gamma\delta} = \delta^\gamma_{[\alpha}\delta^\delta_{\beta]}\,, \tag{A.7}$$

$$C_{\alpha\beta}C^{\alpha\delta} = \delta^\delta_\beta\,. \tag{A.8}$$

For every spinorial field, we define

$$\psi^2 = \frac{1}{2}\psi^\alpha\psi_\alpha = i\psi^+\psi^-\,, \qquad \psi^\alpha\psi_\alpha = -\psi_\alpha\psi^\alpha\,. \tag{A.9}$$

By using the spinor metric, we know that the gamma matrices are symmetric, namely,

$$(\gamma_{\underline{a}})_{\alpha\beta} = (\gamma_{\underline{a}})_{\beta\alpha}\,, \qquad (\gamma_{\underline{a}})^{\alpha\beta} = (\gamma_{\underline{a}})^{\beta\alpha}\,. \tag{A.10}$$

Below, we list some useful identities of gamma matrices.

$$\gamma^{\underline{a}}\gamma_{\underline{a}} = 3\mathbb{I}\,, \tag{A.11}$$

$$\gamma_{\underline{a}}\gamma_{\underline{b}} = -\epsilon_{\underline{abc}}\gamma^{\underline{c}} + \eta_{\underline{ab}}\mathbb{I}\,, \tag{A.12}$$

$$\gamma^{\underline{b}}\gamma_{\underline{a}}\gamma_{\underline{b}} = -\gamma_{\underline{a}}\,, \tag{A.13}$$

$$(\gamma^{\underline{a}})_{\alpha\beta}(\gamma_{\underline{a}})^{\gamma\delta} = -\frac{3}{2}\delta^\gamma_\alpha\delta^\delta_\beta - \frac{1}{2}(\gamma^{\underline{a}})_\alpha{}^\gamma(\gamma_{\underline{a}})_\beta{}^\delta\,, \tag{A.14}$$

$$(\gamma^{\underline{a}})_{\alpha\beta}(\gamma_{\underline{a}})^{\gamma\delta} = -\delta^\gamma_{(\alpha}\delta^\delta_{\beta)} = -(\gamma^{\underline{a}})_{(\alpha}{}^\gamma(\gamma_{\underline{a}})_{\beta)}{}^\delta\,, \tag{A.15}$$

where $\epsilon^{012} = 1$.

**Symmetrization/anti-symmetrization**   The symmetrization of indices is denoted by the round bracket; the antisymmetrization of indices is denoted by the square bracket:

$$A_{(\alpha} B_{\beta)} := A_\alpha B_\beta + A_\beta B_\alpha , \tag{A.16}$$

$$A_{[\alpha} B_{\beta]} := A_\alpha B_\beta - A_\beta B_\alpha = -C_{\alpha\beta} A^\gamma B_\gamma . \tag{A.17}$$

**Covariant derivatives**   The superspace covariant derivatives are defined as $D_A = (\partial_{\alpha\beta}, D_\alpha)$, where

$$\partial_{\alpha\beta} = i (\gamma^{\underline{a}})_{\alpha\beta} \partial_{\underline{a}} ,$$
$$D_\alpha = \partial_\alpha + i \theta^\beta \partial_{\alpha\beta} . \tag{A.18}$$

They satisfy the algebra

$$\{ D_\alpha , D_\beta \} = 2i \partial_{\alpha\beta} ,$$
$$[ \partial_{\alpha\beta} , D_\gamma ] = 0 . \tag{A.19}$$

Below, we list some identities for covariant derivatives, which are useful in the calculations we have encountered throughout this paper.

$$\partial_{\alpha\beta} \partial_\gamma{}^\alpha = C_{\beta\gamma} \Box , \tag{A.20}$$

$$D_\alpha D_\beta = i \partial_{\alpha\beta} - C_{\alpha\beta} D^2 , \tag{A.21}$$

$$D^2 D_\alpha = -D_\alpha D^2 = i \partial_{\alpha\beta} D^\beta , \tag{A.22}$$

$$D^\beta D_\alpha D_\beta = 0 , \tag{A.23}$$

$$(D^2)^2 = \Box , \tag{A.24}$$

$$\partial_{\alpha(\beta} D_{\gamma)} = 2 \partial_{\beta\gamma} D_\alpha + i C_{\alpha(\beta} D_{\gamma)} D^2 , \tag{A.25}$$

$$D^\alpha D_{(\alpha} \partial_{\beta\gamma)} = 8 \partial_{\beta\gamma} D^2 , \tag{A.26}$$

where

$$\Box = \tfrac{1}{2} \partial^{\alpha\beta} \partial_{\alpha\beta} = \partial^{\underline{a}} \partial_{\underline{a}} ,$$
$$D^2 = \tfrac{1}{2} D^\alpha D_\alpha . \tag{A.27}$$

**Map between vector and spinorial representation**   Here we summarize the mapping between vector and spinorial representations.

$$
\begin{aligned}
\text{For fields:} \quad & A_{\underline{a}} = \frac{i}{2} (\gamma_{\underline{a}})^{\alpha\beta} A_{\alpha\beta} , \quad A_{\alpha\beta} = i (\gamma^{\underline{a}})_{\alpha\beta} A_{\underline{a}} , \\
\text{For derivatives:} \quad & \partial_{\underline{a}} = \frac{i}{2} (\gamma_{\underline{a}})^{\alpha\beta} \partial_{\alpha\beta} , \quad \partial_{\alpha\beta} = i (\gamma^{\underline{a}})_{\alpha\beta} \partial_{\underline{a}} , \\
\text{For coordinates:} \quad & x_{\underline{a}} = i (\gamma_{\underline{a}})^{\alpha\beta} x_{\alpha\beta} , \quad x_{\alpha\beta} = \frac{i}{2} (\gamma^{\underline{a}})^{\alpha\beta} x_{\underline{a}} .
\end{aligned}
\tag{A.28}
$$

## A.2   Identities

In this section, we collect identities that we have used in the derivations in section 3.

**Identity 1** The first identity is a consequence of the nilpotency condition for $W_\alpha$. Notice that $W^2 \partial_{\beta\gamma}(W^2 h)$ is identically zero due to $W^3 \equiv 0$, therefore $D^2\left[W^2\partial_{\beta\gamma}(W^2 h)\right]$ is also identically zero.

$$0 = D^2\left[W^2\partial_{\beta\gamma}(W^2 h)\right] = (D^2 W^2)\partial_{\beta\gamma}(W^2 h) + W^2\,\partial_{\beta\gamma}D^2(W^2 h) + (D^\alpha W^2)\partial_{\beta\gamma}D_\alpha\left[W^2 h\right]. \quad (A.29)$$

The second equal sign comes from distributing the covariant derivative D. A simple rewriting yields

$$(D^\alpha W^2)\partial_{\beta\gamma}D_\alpha\left[W^2 h\right] = -(D^2 W^2)\partial_{\beta\gamma}(W^2 h) - W^2\,\partial_{\beta\gamma}D^2(W^2 h), \quad (A.30)$$

which is equivalent to (113).

**Identity 2** The second identity is obtained by the EoM (134) and the property of spinors.

$$\begin{aligned}
\left(D^\alpha U_{(\beta}\,\tilde{g}\right)\left(D_{\gamma)}U_\alpha\right) &= \left(D_{(\beta}U^\alpha\,\tilde{g}\right)\left(D_{\gamma)}U_\alpha\right) \\
&= -U^\alpha\left(D_{(\beta}\tilde{g}\right)\left(D_{\gamma)}U_\alpha\right) \\
&= -\frac{1}{2}\left(D_{(\beta}\tilde{g}\right)\left(D_{\gamma)}U^\alpha U_\alpha\right) = -\left(D_{(\beta}\tilde{g}\right)\left(D_{\gamma)}U^2\right),
\end{aligned} \quad (A.31)$$

where the first line comes directly from the EoM (134); the second line is obtained by $\left(D_{(\beta}U^\alpha\right)\left(D_{\gamma)}U_\alpha\right) \equiv 0$.

**Identity 3** The third identity is an application of integration by parts:

$$\begin{aligned}
2i\,U_{(\beta}\,\tilde{g}\left(\partial_{\gamma)}{}^\delta U_\delta\right) &= 2i\,\partial_{(\gamma}{}^\delta\left(U_{\beta)}\,\tilde{g}U_\delta\right) - 2i\left(\partial_{(\gamma}{}^\delta U_{\beta)}\,\tilde{g}\right)U_\delta \\
&= 4i\,\partial_{\beta\gamma}\left[\tilde{g}\,U^2\right] - 2i\left(\partial_{(\gamma}{}^\delta U_{\beta)}\right)\tilde{g}\,U_\delta - 2i\,U_{(\beta}\left(\partial_{\gamma)}{}^\delta\tilde{g}\right)U_\delta.
\end{aligned} \quad (A.32)$$

**Identity 4** Here, we give a sketch of the proof of (116). We start by considering the following expression

$$\mathcal{I}_{\beta\gamma} := \left(D_\alpha D_{(\beta}W^2\right)\partial_{\gamma)}{}^\alpha\left[W^2 f'(T)\right]. \quad (A.33)$$

1. Using (A.21) in the first factor of $\mathcal{I}_{\beta\gamma}$

$$\begin{aligned}
\mathcal{I}_{\beta\gamma} &= \left(i\,\partial_{\alpha(\beta}W^2\right)\partial_{\gamma)}{}^\alpha\left[W^2 f'(T)\right] - 2(D^2 W^2)\partial_{\beta\gamma}\left[W^2 f'(T)\right] \\
&= \partial_{\alpha(\beta}\left(W^2\partial_{\gamma)}{}^\alpha\left[W^2 f'(T)\right]\right) - W^2\partial_{\alpha(\beta}\partial_{\gamma)}{}^\alpha\left[W^2 f'(T)\right] - 2(D^2 W^2)\partial_{\beta\gamma}\left[W^2 f'(T)\right],
\end{aligned} \quad (A.34)$$

where the first term is zero due to nilpotency and the second term is also zero by symmetry (A.20). Namely,

$$\mathcal{I}_{\beta\gamma} = -2(D^2 W^2)\partial_{\beta\gamma}\left[W^2 f'(T)\right]. \quad (A.35)$$

2. Using (A.21) in the second factor of $\mathcal{I}_{\beta\gamma}$

$$\begin{aligned}
\mathcal{I}_{\beta\gamma} &= i\left(D^\alpha D_{(\beta}W^2\right)D_{|\alpha|}D_{\gamma)}\left[W^2 f'(T)\right] - 2\,\partial_{\beta\gamma}\left(W^2 D^2\left[W^2 f'(T)\right]\right) \\
&\quad + 2\,W^2\,\partial_{\beta\gamma}D^2\left[W^2 f'(T)\right].
\end{aligned} \quad (A.36)$$

Note that the first term can be simplified by applying the following equation,

$$D^2\Big[(D_{(\beta}W^2)(D_{\gamma)}W^2 f'(T))\Big] = \Big[D^2 D_{(\beta}W^2\Big]\Big[D_{\gamma)}W^2 f'(T)\Big] + \Big[D_{(\beta}W^2\Big]\Big[D^2 D_{\gamma)}W^2 f'(T)\Big]$$
$$- \Big(D^\alpha D_{(\beta}W^2\Big)D_{|\alpha|}D_{\gamma)}\Big[W^2 f'(T)\Big]$$
$$= -\Big[D_{(\beta}D^2 W^2\Big]\Big[D_{\gamma)}W^2 f'(T)\Big] - \Big[D_{(\beta}W^2\Big]\Big[D_{\gamma)}D^2 W^2 f'(T)\Big]$$
$$- \Big(D^\alpha D_{(\beta}W^2\Big)D_{|\alpha|}D_{\gamma)}\Big[W^2 f'(T)\Big].$$
(A.37)

The LHS is actually zero,

$$(D_{(\beta}W^2)\big(D_{\gamma)}W^2 f'(T)\big) = (D_{(\beta}W^2)\big(D_{\gamma)}W^2\big)f'(T) + (D_{(\beta}W^2)W^2\big(D_{\gamma)}f'(T)\big), \quad (A.38)$$

where the first term is zero due to the symmetrization of anti-commuting fermions, while the second term vanishes due to nilpotency. So,

$$\mathcal{I}_{\beta\gamma} = -i\Big[D_{(\beta}D^2 W^2\Big]\Big[D_{\gamma)}W^2 f'(T)\Big] - i\Big[D_{(\beta}W^2\Big]\Big[D_{\gamma)}D^2 W^2 f'(T)\Big]$$
$$- 2\,\partial_{\beta\gamma}\Big(W^2 D^2\Big[W^2 f'(T)\Big]\Big) + 2 W^2\,\partial_{\beta\gamma}D^2\Big[W^2 f'(T)\Big].$$
(A.39)

3. Comparing (A.35) and (A.39), we find

$$-2(D^2 W^2)\,\partial_{\beta\gamma}\Big[W^2 f'(T)\Big] = -i\Big[D_{(\beta}D^2 W^2\Big]\Big[D_{\gamma)}W^2 f'(T)\Big] - i\Big[D_{(\beta}W^2\Big]\Big[D_{\gamma)}D^2 W^2 f'(T)\Big]$$
$$- 2\,\partial_{\beta\gamma}\Big(W^2 D^2\Big[W^2 f'(T)\Big]\Big) + 2 W^2\,\partial_{\beta\gamma}D^2\Big[W^2 f'(T)\Big].$$
(A.40)

Using the definition of $g$ (107), after some algebra, we get

$$(D_{(\beta}W^2)(D_{\gamma)}g) = -2i\,W^2\,\partial_{\beta\gamma}g + 4i\,W^2\,\partial_{\beta\gamma}f(T) - 2\big(D_{(\beta}f(T)\big)\big(D_{\gamma)}W^2\big). \quad (A.41)$$

## A.3 Off-shell expressions for supercurrent-squared operators in TG

In this section, we present the off-shell expressions for the supercurrent-squared operators $\{\mathcal{O}_{T^2}, \mathcal{O}_{\Theta^2}, \mathcal{O}_{\Theta R}\}$ in the tensor-Goldstone multiplet. As mentioned in the main text, the on-shell expressions (142) can be computed using similar arguments as those in section 3.3 due to the property (137). Namely, with the support of $U^2\tilde{g}$, the equations of motion yield $D_{(\alpha}U_{\beta)} \approx 2 D_\alpha U_\beta$. To derive the off-shell results, we first use the following identity

$$D_{(\alpha}U_{\beta)} = 2 D_\alpha U_\beta - D_{[\alpha}U_{\beta]} = 2 D_\alpha U_\beta + C_{\alpha\beta}\mathcal{H}, \quad (A.42)$$

where (A.17) is used and we define

$$\mathcal{H} := D^\gamma U_\gamma. \quad (A.43)$$

Note that the auxiliary field $H$ is the lowest component of the superfield $\mathcal{H}$: $H = \mathcal{H}|$. Equation (137) is equivalent to $U^2\tilde{g}\,\mathcal{H} = 0$. After some algebra, we get

$$\mathcal{O}_{T^2} = -4!2\,\tilde{\kappa}^2\,U^2\,\tilde{T}\Big[h(\tilde{T}) + \tilde{T}\,h'(\tilde{T})\Big]^2 - 4!\,U^2\,\tilde{g}^2\,\mathcal{H}^2, \quad (A.44a)$$

$$\mathcal{O}_{\Theta^2} = \frac{\tilde{\kappa}^2}{2}\,\tilde{T}\,U^2\Big[h(\tilde{T}) - 2\,\tilde{T}\,h'(\tilde{T})\Big]^2, \quad (A.44b)$$

$$\mathcal{O}_{\Theta R} = 4!2\,\sqrt{6}\,\tilde{\kappa}^2\,U^2\,\tilde{T}\Big[h(\tilde{T}) - 2\,\tilde{T}\,h'(\tilde{T})\Big]\Big[h(\tilde{T}) + \tilde{T}\,h'(\tilde{T})\Big] \quad (A.44c)$$

$$+ U^2\Big[h(\tilde{T}) - 2\,\tilde{T}\,h'(\tilde{T})\Big]\frac{\tilde{g}^2\,\mathcal{H}\,p(\mathcal{H})}{4!4\sqrt{6}\,\tilde{\kappa}^2\,U^2\,\tilde{T}\Big[h(\tilde{T}) + \tilde{T}\,h'(\tilde{T})\Big]} + \cdots,$$

where

$$p(\mathcal{H}) = 3!3!2\left[\mathcal{H}^3 + 4\tilde{\kappa}^2\tilde{T}\,\mathcal{H} - (D_\delta U^\gamma)(D_\gamma U^\delta)\mathcal{H} - 2(D_\delta U^\beta)(D_\beta U^\gamma)(D_\gamma U^\delta)\right], \qquad \text{(A.45)}$$

and the $\cdots$ in the end of (A.44c) denotes terms which are higher order in $\tilde{g}^2\,\mathcal{H}\,p(\mathcal{H})$. Clearly, the second term in $\mathcal{O}_{T^2}$ and the second line in $\mathcal{O}_{\Theta R}$ vanish on-shell and (A.44) reduces to (142).

# B  Conventions for sections 2 and 4

## B.1  Conventions

**Indices**   In sections 2 and 4 we used Latin letters to denote spacetime indices: $a = 0, 1, 2$; and we use Greek letters for spinorial indices: $\alpha = 1, 2$.

**Gamma matrix**   We choose $3d$ gamma matrices as

$$(\gamma^0)_\alpha{}^\beta = i(\sigma^2)_\alpha{}^\beta, \qquad (\gamma^1)_\alpha{}^\beta = (\sigma^3)_\alpha{}^\beta, \qquad (\gamma^2)_\alpha{}^\beta = -(\sigma^1)_\alpha{}^\beta, \qquad \text{(B.1)}$$

which satisfy:

$$\{\gamma^a, \gamma^b\} = 2\eta^{ab}\mathbb{I}, \qquad (\gamma_a)_\alpha{}^\gamma(\gamma_b)_\gamma{}^\beta = \eta_{ab}\delta_\alpha^\beta + \varepsilon_{abc}(\gamma^c)_\alpha{}^\beta, \qquad \text{(B.2)}$$

where the Minkowski metric and the Levi-Civita tensors are

$$\eta_{ab} = \eta^{ab} = \text{diag}(-1, 1, 1), \qquad \varepsilon_{012} = -1, \quad \varepsilon^{012} = 1. \qquad \text{(B.3)}$$

Spinor indices are raised and lowered as follows:

$$\psi_\alpha = \varepsilon_{\alpha\beta}\psi^\beta, \qquad \psi^\alpha = \varepsilon^{\alpha\beta}\psi_\beta, \qquad \text{(B.4)}$$

where

$$\varepsilon^{\alpha\beta} = -\varepsilon^{\beta\alpha}, \qquad \varepsilon_{\alpha\beta} = -\varepsilon_{\beta\alpha}, \qquad \varepsilon^{12} = \varepsilon_{21} = 1, \qquad \text{(B.5)}$$

such that

$$\varepsilon_{\alpha\beta}\varepsilon^{\gamma\delta} = -2\delta^\gamma_{[\alpha}\delta^\delta_{\beta]}, \qquad \varepsilon_{\beta\alpha}\varepsilon^{\alpha\delta} = \delta^\delta_\beta. \qquad \text{(B.6)}$$

We use the spinor contraction

$$\psi^2 = \psi^\alpha\psi_\alpha = -\psi_\alpha\psi^\alpha. \qquad \text{(B.7)}$$

The symmetric gamma matrices with up and down indices are

$$(\gamma_a)_{\alpha\beta} = (\gamma_a)_{\beta\alpha} := \varepsilon_{\beta\gamma}(\gamma_a)_\alpha{}^\gamma, \qquad (\gamma_a)^{\alpha\beta} = (\gamma_a)^{\beta\alpha} := \varepsilon^{\alpha\gamma}(\gamma_a)_\alpha{}^\beta. \qquad \text{(B.8)}$$

**Symmetrization/anti-symmetrization**   The symmetrization of indices is denoted by the round bracket; the antisymmetrization of indices is denoted by the square bracket:

$$A_{(\alpha}B_{\beta)} := \frac{1}{2}\left(A_\alpha B_\beta + A_\beta B_\alpha\right), \qquad \text{(B.9)}$$

$$A_{[\alpha}B_{\beta]} := \frac{1}{2}\left(A_\alpha B_\beta - A_\beta B_\alpha\right). \qquad \text{(B.10)}$$

Similarly for (anti-)symmetrization of vector indices. For $n$ indices, (anti-)symmetrization includes an implicit factor of $1/n!$.

**Covariant Derivatives**  The superspace coordinates are defined as $x^A = (x^a, \theta^a)$ with $\theta^\alpha$ being a real Grassmann coordinate. The flat superspace covariant derivatives are defined as $D_A = (\partial_a, D_\alpha)$, where

$$D_\alpha = \partial_\alpha + i\,\theta^\beta\,\partial_{\alpha\beta}\,, \quad \partial_\alpha := \frac{\partial}{\partial\theta^\alpha}\,, \quad \partial_{\alpha\beta} = (\gamma^a)_{\alpha\beta}\partial_a\,. \tag{B.11}$$

They satisfy the algebra

$$\{D_\alpha, D_\beta\} = 2i\,\partial_{\alpha\beta}\,, \quad [\partial_a, D_\beta] = 0\,. \tag{B.12}$$

The operators $D^2$ and $\Box$ are defined as

$$\Box := \partial^a\,\partial_a = -\frac{1}{2}\partial^{\alpha\beta}\partial_{\alpha\beta}\,, \quad D^2 = D^\alpha D_\alpha\,. \tag{B.13}$$

Note that given a vector index, whether used for a field, a coordinate, or a derivative, the map between vector and symmetric bi-spinors is the same. For instance, given a vector $V_a$ we define the symmetric bi-spinor $V_{\alpha\beta}$ by

$$V_{\alpha\beta} := (\gamma^a)_{\alpha\beta}V_a\,, \quad V_a = -\frac{1}{2}(\gamma_a)^{\alpha\beta}V_{\alpha\beta}\,. \tag{B.14}$$

## B.2   Map of conventions between sections 3 and 4

To conclude this Appendix, we provide rules to map results between Section 4 and Section 3. The following correspondences hold where the left-hand side in each pair of columns refers to notation used in Section 4, while the right-hand side of each pair refers to Section 3 and Appendix A. One can map results in Sections 4 to results in Section 3, or vice-versa, by replacing all instances of the symbols in one column with the corresponding symbols in the other column of the pair. Note in particular that this table does not give *equalities* between pairs of symbols, but rather *replacement rules* which convert an expression from one set of conventions to another.

Note also that given a vector field, similarly to the vector derivative, the map of notation is not only a replacement rule but an equivalence:

Section 2 follows the conventions of Section 4, although the only entries of Table 2 and Table 3 used in Section 2 are those for vector indices $V_a$ (which are not underlined) and the conversion between vectors indices and pairs of spinor indices.

Table 2: A set of replacement rules for converting between the notation of Section 4 and that of Section 3.

| Section 4 | Section 3 | Section 4 | Section 3 | Section 4 | Section 3 |
|---|---|---|---|---|---|
| $(\gamma^a)_\alpha{}^\beta$ | $(\gamma^{\underline{a}})_\alpha{}^\beta$ | $\varepsilon_{abc}$ | $-\epsilon_{\underline{abc}}$ | $(\gamma^a)^{\alpha\beta}$ | $i(\gamma^{\underline{a}})^{\alpha\beta}$ |
| $(\gamma^a)_{\alpha\beta}$ | $i(\gamma^{\underline{a}})_{\alpha\beta}$ | $\psi^2$ | $2i\psi^2$ | $\psi_\alpha$ | $\psi_\alpha$ |
| $\psi^\alpha$ | $i\psi^\alpha$ | $\theta^\alpha$ | $\theta^\alpha$ | $\theta_\alpha$ | $-i\theta_\alpha$ |
| $\partial_{\alpha\beta}$ | $\partial_{\alpha\beta}$ | $D_\alpha = \partial_\alpha + i\theta^\beta\partial_{\alpha\beta}$ | $D_\alpha = \partial_\alpha + i\theta^\beta\partial_{\alpha\beta}$ | $D^\alpha$ | $iD^\alpha$ |
| $D^2$ | $2iD^2$ | $\{D_\alpha, D_\beta\} = 2i\partial_{\alpha\beta}$ | $\{D_\alpha, D_\beta\} = 2i\partial_{\alpha\beta}$ | $\Box$ | $\Box$ |
| $A_{(\alpha_1\cdots\alpha_n)}$ | $\frac{1}{n!}A_{(\alpha_1\cdots\alpha_n)}$ | $A_{[\alpha_1\cdots\alpha_n]}$ | $\frac{1}{n!}A_{[\alpha_1\cdots\alpha_n]}$ | $\varepsilon^{\alpha\beta}$ | $iC^{\alpha\beta}$ |

Table 3: The maps of notation between vector indices and pairs of spinor indices in Section 4 and Section 3 are equalities.

| Section 4 | Section 3 | Section 4 | Section 3 |
|---|---|---|---|
| $V_a$ | $V_{\underline{a}}$ | $V_{\alpha\beta}$ | $V_{\alpha\beta}$ |

# C  Off-shell flow for bosonic tensor-Goldstone Lagrangian

In this Appendix we will present a modified version of the flow considered in Section 2.2 which holds fully off-shell. The modification relies upon a reformulation of the Lagrangian's dependence on the auxiliary field $H$ which makes its coupling to the metric similar to that of the scalar field kinetic term $\partial^a \phi \partial_a \phi$.

Recall that the initial condition for the bosonic tensor-Goldstone flow equation is

$$\mathcal{L}_0 = -\partial^a \phi \partial_a \phi + H^2 \,. \tag{C.1}$$

Here $H^2$ is a Lorentz scalar which is completely independent of the metric. However, one could rewrite the theory in a different way as follows. We introduce a vector field $v^a$ with the properties that

$$v^a v_a = 1 \,, \qquad v^a v^b \partial_a \phi \partial_b \phi = \partial^a \phi \partial_a \phi \,. \tag{C.2}$$

The vector is not a new dynamical degree of freedom, but rather plays a role which is somewhat similar to the auxiliary scalar field $a$ in the PST formalism [106–108] whose gradient $v_m = \partial_m a$ is also normalized to unit length. In our case, the presence of the vector field $v^a$ will not affect any of the on-shell dynamics of the theory but is merely a trick which modifies the coupling of the $H^2$ term to a background metric.

We now define the two invariant combinations

$$x_1 = \partial^a \phi \partial_a \phi \,, \qquad x_2 = (Hv^a)(Hv_a) = H^2 v^a v_a = H^2 \,. \tag{C.3}$$

We emphasize that, due to the normalization condition $v^a v_a = 1$, these variables are identical to those used in Section 2.2 when considering theories on a fixed background metric; all that has changed is the coupling to metric fluctuations, and therefore the Hilbert stress tensor. For a general Lagrangian $\mathcal{L}(x_1, x_2)$, one now finds

$$T_{ab} = \eta_{ab} \mathcal{L} - 2 \frac{\partial \mathcal{L}}{\partial g^{ab}} = \eta_{ab} \mathcal{L} - 2 \frac{\partial \mathcal{L}}{\partial x_1} \partial_a \phi \partial_b \phi - 2 \frac{\partial \mathcal{L}}{\partial x_2} H^2 v_a v_b \,, \tag{C.4}$$

and the contractions of the stress tensor which we will need are

$$T^{ab} T_{ab} = 3\mathcal{L}^2 - 4 x_1 \mathcal{L} \frac{\partial \mathcal{L}}{\partial x_1} - 4 \mathcal{L} x_2 \frac{\partial \mathcal{L}}{\partial x_2} + 4 x_1^2 \left( \frac{\partial \mathcal{L}}{\partial x_1} \right)^2 + 8 x_1 x_2 \frac{\partial \mathcal{L}}{\partial x_1} \frac{\partial \mathcal{L}}{\partial x_2} + 4 x_2^2 \left( \frac{\partial \mathcal{L}}{\partial x_2} \right)^2 \,,$$

$$T^a{}_a = 3\mathcal{L} - 2 x_1 \frac{\partial \mathcal{L}}{\partial x_1} - 2 x_2 \frac{\partial \mathcal{L}}{\partial x_2} \,. \tag{C.5}$$

Note that the factor of $(v^a \partial_a \phi)^2$ in the cross-term of $T^{ab} T_{ab}$ collapses to $x_1 x_2$ by the second assumption of (C.2). The root-$T\bar{T}$-like combination is now

$$R = \sqrt{\frac{3}{8} \left( T^{ab} T_{ab} - \frac{1}{3} \left( T^a{}_a \right)^2 \right)} = \sqrt{\left( x_1 \frac{\partial \mathcal{L}}{\partial x_1} + x_2 \frac{\partial \mathcal{L}}{\partial x_2} \right)^2} \,, \tag{C.6}$$

and if we assume that

$$x_1 \frac{\partial \mathcal{L}}{\partial x_1} + x_2 \frac{\partial \mathcal{L}}{\partial x_2} > 0 \,, \tag{C.7}$$

then we can take the square root to write

$$R = x_1 \frac{\partial \mathcal{L}}{\partial x_1} + x_2 \frac{\partial \mathcal{L}}{\partial x_2} \,. \tag{C.8}$$

Now the flow equation becomes

$$\frac{\partial \mathcal{L}}{\partial \lambda} = \frac{1}{6} T^{ab} T_{ab} - \frac{1}{9} \left( T^a_{\ a} \right)^2 + \frac{1}{9} \left( T^a_{\ a} \right) R$$
$$= -\frac{1}{2} \mathcal{L}^2 + \mathcal{L} \left( x_1 \frac{\partial \mathcal{L}}{\partial x_1} + x_2 \frac{\partial \mathcal{L}}{\partial x_2} \right), \tag{C.9}$$

and the solution to this differential equation with initial condition $\mathcal{L}_0 = -x_1 + x_2$ is

$$\mathcal{L}(\lambda) = \frac{1}{\lambda} \left( 1 - \sqrt{1 + 2\lambda(x_1 + x_2)} \right). \tag{C.10}$$

This is a fully off-shell solution to the flow equation which does not require assuming $H = 0$ in any intermediate steps. The upshot of this simple exercise is that a trivial modification to the way in which the auxiliary field $H$ enters the Lagrangian can change the properties of the off-shell solution to the flow equation. It would be interesting to understand whether this argument can be supersymmetrized, which would lead to an off-shell version of the superspace flow equation for the tensor-Goldstone multiplet presented in Section 3.

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
