# Peer review of "$T \overline{T}$-Like Flows and $3d$ Nonlinear Supersymmetry"

_SciPost Physics, doi:SciPost Phys. 16, 038 (2024)_

## Round 3 · Referee Report · Anonymous (Referee 1) · 2023-11-12

Strengths

1- The authors performed quite a bit of technical calculations, which they present clearly and well-framed with physical argumentation.

2- This paper extends the applicability of "stress tensor squared" type deformations to 3 dimensional theories, with and without supersymmetry. These theories describe known string theory objects and are related to non-linearly realized supersymmetries.

3- The authors point out that flows closely related to these irrelevant deformations can be rewritten as relevant deformations (albeit somewhat complicated ones) as some of them recently discovered. This may offer an avenue to understanding what makes these irrelevant deformations special.

Weaknesses

1- This type of flow between different theories has proven useful for the calculation of partition functions, energy spectra and S-matrices. Whereas this paper constructs the flows in 3d (SUSY) (Dirac)-Born-Infeld theories, there are no comments on how these flows can be used to simplify such calculations.

2- Another important property of these integrable flows, like the $T \overline{T}$ one, is that they are well-defined by point splitting in the quantum theory. The authors do not comment on whether this holds in this case, or whether there is another property that singles out these particular operators at the quantum level.

Report

In this paper the authors show that the 3d Born-Infeld and Dirac-Born-Infeld theories can be obtained from a "stress tensor squared" type flow. They also show that the $\mathcal{N} = 1$ supersymmetric extension, the Maxwell-Goldstone and tensor-Goldstone models, are similarly generated by squares of the supercurrents, which they derive both indirectly from consistency and conservation as well as directly from linear coupling to supergravity. These theories are relevant to the description of D-branes and are known to have a additional, spontaneously broken supersymmetry, which the authors suspect might be related to the fact that these theories can be obtained by supercurrent squared type deformations.

These are new and interesting results which raise some further interesting questions. There are some concrete issues listed in the "requested changes" paragraph that the authors should address. The paper does belong in SciPost Physics and, when these issues are addressed, will meet the expectations and criteria for this journal.

Beyond these concrete issues, I would also warmly welcome any comments on the more general observations in the "weaknesses" paragraph of this report.

Requested changes

1- Near equations (2.10) and (2.39) the authors make a specific sign choice for the square root of an operator. Does this mean only a part of the phase space of the theory gets deformed? If we continue the other part of the phase space with the same deformation but the other sign of the square root, does the deformed phase space retain any kind of continuity across the threshold value of $\mathfrak{t}$ and $x_1$? If not, how are we supposed to interpret the resulting theory? Furthermore, this issue wasn’t discussed explicitly in the supersymmetric section, e.g. around equation (3.41). Is the situation somehow better in the SUSY theory or is it the same? 2- In going from (2.18) to (2.19), the authors disregard the possibility of total derivative terms contributing. However below (2.58) such terms are deemed essential. One of these arguments should be explained more clearly so it is obvious why the same argument does not carry over to the other case. At the quantum level, the presence of an improvement term is part of the definition of the theory, for example in $d = 2$ it can change the central charge. Would this invalidate the derivation around (2.18)? 3- It is extremely interesting that the flow between "subtracted Lagrangians" can be massaged into an RG-relevant one. However this does raise some questions regarding the unusual properties (no UV CFT behavior, non-locality, etc.) of $T \overline{T}$ type deformations, which are not expected from relevant deformations. Are these properties due to the non-analytic form of the relevant operator? Any other comments? Thanks.

  • validity: high
  • significance: good
  • originality: good
  • clarity: high
  • formatting: excellent
  • grammar: excellent

Author:  Christian Ferko  on 2023-12-14  [id 4192]

(in reply to Report 1 on 2023-11-12)

We thank the referee for reading our manuscript and for his or her comments.

Let us first address the two points raised in the ``Weaknesses'' section of the referee report, which are in fact related. All of the observables which the referee mentioned (namely partition functions, energy spectra and S-matrices) are defined in the quantum theory, and therefore one would first need to define a quantum-mechanical version of our deforming operators in order to derive flow equations for any of these quantities. The issue of defining such a quantum operator is raised in the referee's second point. We should mention that the purely bosonic combination introduced in our equation (2.14) does not give rise to a well-defined operator by point-splitting, even in a conformal field theory. The reason for this is that the trace $T^a_{\; \; a}$ vanishes in such a theory, so only the first term in our deforming operator survives, while in a $d$-dimensional conformal field theory one has

\begin{align}
\langle T_{ab} ( x ) T^{ab} ( 0 ) \rangle = \frac{d - 2}{x^{2d}} + \text{regular} \, .
\end{align}

The first term only vanishes in $d = 2$ but not in any higher dimension. Because of this position dependence, one cannot take the limit $x \to 0$ to define a local operator in $d = 3$, as one does in two dimensions.

Although defining a quantum $T \overline{T}$-like operator in $3d$ is outside of the scope of this paper, one might speculate that introducing supersymmetry might be necessary in order to do so. The reason for this is that supersymmetric extensions of the $T \overline{T}$ operator, such as those considered in our Section 3, contain contributions from additional fields in the stress tensor multiplet. One might hope that, in theories with sufficient supersymmetry, a certain combination which includes both $T_{ab} T^{ab}$ and also additional terms involving these extra fields in the multiplet might give rise to a position-independent two-point function that could then be used to define a local operator by point-splitting. Again, we did not consider this question in the present work, but we feel that an analysis of supersymmetric extensions of the $T \overline{T}$ operator in $d > 2$ such as the one in this manuscript might be a necessary first step towards solving this problem.

In response to the three items in the ``Requested changes'' section of the report:

(1) Although we did not comment on it explicitly in the supersymmetric discussion, the same sign choice for $\mathfrak{t}$ or $x_1$ appears in this setting, since truncating these supersymmetric flows to the bosonic sector reproduces the flows of Section 2, and therefore all of the same sign choices must apply.

As the referee points out, strictly speaking, we have only explained how to deform part of the phase space (e.g. $| \vec{E} |^2 < | \vec{B} |^2)$ of the Maxwell theory into the Born-Infeld theory. However, it is straightforward to deform the other part of the phase space by choosing slightly different coefficients in the definition of our deforming operator (as we briefly mention below equation (2.11)).

One could therefore consider a piecewise-defined version of the deformation, which acts differently in the two parts of the phase space, to handle both cases at once. Note in particular that this piecewise definition is continuous across the transition point, where $\mathfrak{t} = 0$ and therefore the operator $R$ vanishes, so either choice of $\pm R$ is equivalent.

We have added a sentence below equation (2.11) in the revised manuscript to clarify this point.

(2) We have clarified the discussion around equation (2.18) and added comments about how it relates to equation (2.58). What we meant to emphasize is that equations (2.19) and (2.20) only hold up to total derivative ambiguities; we did not mean to suggest that these can always be ignored, since as we see around equation (2.58), these total derivative terms can be important in some contexts.

We agree with the referee that, in the quantum theory, the spectrum of local operators (including the stress tensor) is part of the definition of the theory, and there is no freedom to perform improvement transformations. However, although it is outside the scope of the present work, it is possible to argue that the trace flow equation for the $T \overline{T}$ operator holds at the quantum level, without resorting to the classical argument we have presented that suffers from total derivative ambiguities. (For instance, one way that this trace flow equation has been justified in the literature is by performing a gravity analysis in the holographic dual to a $T \overline{T}$ deformed CFT.)

(3) We agree that this rewriting of the flow in terms of a relevant operator is interesting, and we note that the same rewriting can be done for the $2d$ deformation that produces the Nambu-Goto action, as pointed out in 2301.10411. In the $2d$ setting, the irrelevant form of the deformation is of course well-defined at the quantum level, although the interpretation of the relevant form is still not clear.

Although a relevant deformation is not expected to lead to the phenomena observed under a $2d$ $T \overline{T}$ deformation, as the referee mentioned, we should comment that the form of this relevant operator is quite complicated and involves division by a non-analytic combination of stress tensors. Even if one expands around a constant (non-zero) stress tensor background in order to Taylor expand this combination, it leads to an infinite series of stress tensor operators. This is a very complicated object from the perspective of the low energy (undeformed) theory, but it might be more natural if it could be expressed in terms of other objects in the finite-$\lambda$ deformed theory, which is believed to be non-local and therefore does not possess local operators.

One might be tempted to interpret the possibility of performing this RG-relevant rewriting as a signal of UV/IR mixing. Such mixing is a hallmark of non-commutative field theories, and various authors have speculated that a $T \overline{T}$-deformed $2d$ QFT might share some properties with a non-commutative field theory (for instance, both $T \overline{T}$ and non-commutativity lead to CDD-factor-like modifications of the S-matrix). Unfortunately, we do not yet have any precise statements to make on this subject.

We believe that these modifications have improved the overall quality of the paper, and we again thank the referee for these suggestions.

Sincerely,

Christian Ferko, Yangrui Hu, Zejun Huang, Konstantinos Koutrolikos, and Gabriele Tartaglino-Mazzucchelli

---

## Round 3 · Referee Report · Anonymous (Referee 2) · 2023-12-7

Strengths

  • The paper discusses TTbar-like flow equations in D>2 dimensions, which are a challenging and interesting topic
  • The paper finds a flow-equation description for the (Dirac-)Born-Infeld action, as well as for its supersymmetric version
  • The paper is clearly written

Weaknesses

  • The entire discussion of the paper is restricted to the classical action / Lagrangian

Report

This article discusses how the action of three-dimensional Born-Infeld theory, Dirac-Born-Infeld, and supersymmetrisations thereof solve certain flow equations. These equations are similar to the famous TTbar flow equation in D=2, but rather more involved. The paper is clear and well-written.

The "original" D=2 TTbar operator and the resulting flow has many intriguing properties. Firstly, and this is what the authors reproduce, it yields the gauge-fixed string action when applied to free scalars. Additionally, it is well-defined at the quantum level, it acts in a simple way (diagonally) on the S-matrix of the theory, and it preserves several symmetries including integrability (in fact, its effect on the spectrum is solvable in terms of a simple ODE on the energy levels).

Here the authors only focus on the classical action of the model. Moreover, in light of recent results by Seibold and Tseytlin, which indicate that the 3D brane action is not integrable (and that its S-matrix is not diagonal, nor elastic), it seems that the the deformation considered by the authors would necessarily break integrability.

Requested changes

The authors should clarify the properties of their flow equation with respect to integrability and to the S-matrix of the theory (at least at tree-level), especially in relation to the results of Seibold and Tseytlin.

I have also spotted a misprint in eq. (2.34).

  • validity: high
  • significance: high
  • originality: high
  • clarity: high
  • formatting: excellent
  • grammar: perfect

Author:  Christian Ferko  on 2023-12-14  [id 4191]

(in reply to Report 2 on 2023-12-07)

We are grateful to the anonymous referee for giving feedback on our work.

We agree with the reviewer's point that the analysis of our paper is entirely classical, which was also commented upon in the first referee report. It would indeed be quite intriguing if one could extend some of our results to the quantum theory. For instance, it could be interesting to compute the partition functions of the $3d$ theories considered in this work on $\mathbb{R} \times S^2$ or $S^1 \times S^2$ and see whether our deformations are solvable at the quantum level. If one could extend the flows in our paper from $\mathcal{N} = 1$ to $\mathcal{N} = 2$ supersymmetry, it may even be possible to apply localization techniques in this setting.

However, we feel that considering classical flow equations can already serve as a useful first step, especially given that classical aspects of the more well-known two-dimensional $T \overline{T}$ deformation -- such as the fact that it yields the gauge-fixed Nambu-Goto string action, or that solutions to the deformed classical equations of motion can be generated from a field-dependent diffeomorphism -- exhibit a rich structure that has been investigated in many works.

Below we have included more specific responses to the two suggested improvements.

(1) We have added two paragraphs in Section 2.2, at the bottom of page 18 and top of page 19, commenting on the issue of S-matrix integrability and the relationship between our results and those of 2308.12189.

This connection is very interesting; since our $3d$ operator produces the $3d$ membrane action, and the results of Seibold and Tseytlin show that the dimensional reduction of this theory to $2d$ does not have an integrable S-matrix (even at tree level), it indeed follows that the dimensional reduction of our $3d$ operator cannot preserve integrability. However, this fact does not lead to any contradiction, since the dimensional reduction of our $3d$ operator necessarily differs from the usual two-dimensional $T \overline{T}$ operator, which does preserve integrability.

(2) The missing $\phi$ in equation (2.34) has been corrected.

We thank the referee for these remarks, especially for pointing out 2308.12189, and we hope that the revised version of our manuscript will now be suitable for publication in SciPost Physics.

Sincerely,

Christian Ferko, Yangrui Hu, Zejun Huang, Konstantinos Koutrolikos, and Gabriele Tartaglino-Mazzucchelli

---

## Round 4 · Referee Report · Anonymous · 2023-12-20

Report

I would like to recommend the paper for publication and thank the authors for addressing all of my remarks.

---

## Round 4 · Referee Report · Anonymous · 2024-1-1

Report

The authors have addressed my comments and therefore I am happy to recommend the article for publication.

---

## Round 4 · Author Response

Attached is a revised version of our manuscript which incorporates the suggested changes which we received in the refereeing round.

---

## Round 4 · List of Changes

(1) We have added a sentence in the paragraph following equation (2.11) to clarify that we have made a sign choice which is appropriate for deforming field configurations with $\mathfrak{t} < 0$, but that one could have defined a piecewise deformation which works for all $\mathfrak{t}$.

(2) We added two paragraphs, at the bottom of page 18 and top of page 19, explaining the relationship between our results and those of 2308.12189.

(3) We clarified the discussion around equation (2.19) to better explain the role of possible total derivative terms, which are important for the later observations around equation (2.58).

(4) We corrected a typo in equation (2.34).

---

## Editorial Decision

published